# Theoretical Challenges in Learning for Branch-and-Cut

**Hongyu Cheng** [1]  **Amitabh Basu** [1]

## Abstract

Machine learning is increasingly used to guide branch-and-cut (B&C) for mixed-integer linear programming by learning score-based policies for selecting branching variables and cutting planes. Many approaches train on local signals from lookahead heuristics such as strong branching, and linear programming (LP) bound improvement for cut selection. Training and evaluation of the learned models often focus on local score accuracy. We show that such local score-based methods can lead to search trees exponentially larger than optimal tree sizes, by identifying two sources of this gap. The first is that these widely used expert signals can be misaligned with overall tree size. LP bound improvement can select a root cut set that yields an exponentially larger strong branching tree than selecting cuts by a simple proxy score, and strong branching itself can be exponentially suboptimal (Dey et al., 2024). The second is that small discrepancies can be amplified by the branch-and-bound recursion. An arbitrarily small perturbation of the right-hand sides in a root cut set can change the minimum tree size from a single node to exponentially many. For branching, arbitrarily small score discrepancies, and differences only in tie-breaking, can produce trees of exponentially different sizes, and even a small number of decision differences along a trajectory can incur exponential growth. These results show that branch-and-cut policies trained and learned using local expert scores do not guarantee small trees, thus motivating the study of data-driven methods that produce policies better aligned with tree size rather than only accuracy on expert scores.

[1]Department of Applied Mathematics and Statistics, Johns Hopkins University, Baltimore, MD 21218, USA. Correspondence to: Hongyu Cheng <hongyucheng@jhu.edu>, Amitabh Basu <basu.amitabh@jhu.edu>.

*Proceedings of the 43$^{rd}$ International Conference on Machine Learning*, Seoul, South Korea. PMLR 306, 2026. Copyright 2026 by the author(s).

## 1. Introduction

Modern solvers for mixed-integer linear programs (MILPs) rely on branch-and-cut (B&C), which combines branching on integer variables with the addition of cutting planes (Land & Doig, 2009; Nemhauser & Wolsey, 1988; Conforti et al., 2014). A B&C run is driven by a long sequence of decisions, such as which cuts to add, which variable to branch on, and which node to process, while performance is measured by a global criterion, namely the size of the resulting search tree. The problem of making these decisions, a.k.a. *selecting branch-and-cut policies*, is challenging because the overall tree size is a complicated function of these local decisions made at individual nodes of the tree. In principle, one could solve this using dynamic programming, but the enormous size of the state space renders such approaches impractical. Instead, state of the art methods employ policies based on local parameters, for example LP lookahead for cut selection or strong branching scores for branching, meaning that the decision that maximizes such a local parameter is selected at every stage of the branch-and-cut procedure. However, calculating even these local parameters can be computationally very intensive.

**Machine learning for branch-and-cut.** This has motivated a growing literature that uses machine learning to guide B&C decisions, with the goal of reducing tree size and solve time. See (Bengio et al., 2021; Scavuzzo et al., 2024) for surveys and Section 2 below for a broader discussion. Although learned policies often outperform default heuristics on targeted distributions, it remains unclear when their local training objectives translate into improved global B&C performance. Most learning pipelines train on *local supervision*, either by regressing an expert score, learning a ranking over candidate actions, or matching an expert decision via classification. Performance, however, is measured by *global* outcomes, such as the size of the B&C tree or end-to-end solve time. This mismatch leads to a basic question:

> *If a learned policy matches an expert's local scores (or decisions) to high accuracy, does it necessarily have similar global B&C performance?*

There are at least four reasons to be skeptical. First, branch-

and-bound is a sequential decision process, so imitation learning faces a distribution mismatch as a general learning paradigm: training examples come from the node distribution induced by the expert, but at test time the learned policy induces its own distribution. Even a small imitation error rate under the expert distribution can compound along a long trajectory and, in the worst case, yield a quadratic dependence of the cumulative loss on the effective horizon (Ross et al., 2011). Second, at a fixed node, score-based rules select an $\arg\max$ over candidates. Thus, even when score error margins are small or ties occur, tiny score perturbations or different tie-breaking can flip the selected action, and small changes in scoring parameters can lead to very different tree sizes (Balcan et al., 2018). While these two issues are standard imitation learning concerns, our results identify failure modes that persist even in the infinite data limit and under very strong forms of local agreement. In particular, a major cause for concern is that the learned policies have (assuming good generalization) similar expected scores as the expected expert score, but the expectation is with respect to a *distribution over nodes*, since the expert score is local and the training sample comes from such a distribution over nodes. However, we would like the learned policies to have low expected *tree size* with respect to a *distribution over instances* of mixed-integer problems. There can be a severe disconnect because of this difference in the loss function and distribution used for learning, and the loss function and distribution one would like to perform well with respect to. This, combined with the first two issues pointed out above arising from small errors compounding over the overall B&C run, can lead to poor performance of the learned policies in comparison to optimal policies for the given distribution over instances. Finally, standard expert policies themselves may be substantially suboptimal in comparison to optimal policies for the given distribution over instances.

By formally quantifying the above issues, this paper answers the above question in the negative: local imitation accuracy does not control global tree size. In contrast to generalization analyses, our focus is whether local imitation objectives are stable surrogates for global tree size. We isolate two obstacles. The first is *expert suboptimality*: the expert signal used for supervision can itself be far from optimal, so faithfully imitating it can hurt. The second is *perturbation instability*: even when the intended signal is sensible, tiny perturbations in scores or cut definitions can be amplified by the recursion, leading to exponentially different trees. Our results are worst case and do not contradict empirical success on structured instance distributions; their purpose is to clarify the pitfalls in local supervision alone, and to motivate training and evaluation procedures that account for stability.

*Table 1.* Summary of main results. Each entry illustrates a failure mode of local supervision for B&C decisions.

| | **Expert suboptimality** | **Perturbation instability** |
|---|---|---|
| **Cut selection** | Theorem 4.1: LP improvement based selection $\Rightarrow 2^{\Omega(n)}$ larger tree vs. selecting cuts by efficacy. | Theorem 4.3: $\varepsilon$ RHS change in a root cut set $\Rightarrow$ tree size 1 vs. $2^{\Omega(n)}$. |
| **Branching variable selection** | Dey et al. (2024): Strong branching can incur $2^{\Omega(n)}$ blowup vs. optimal. | Theorem 5.1: arbitrarily small score differences $\Rightarrow 2^{\Omega(n)}$ blowup. Theorem 5.4: $k$ deviations from strong branching $\Rightarrow 2^{\Omega(k)}$ blowup. |

**Our contributions.** We establish four separations that demonstrate these failures for cut selection and branching. Table 1 summarizes our results.

For *expert suboptimality*, we show that LP bound improvement, a natural lookahead signal for cut selection, can select a root cut set that yields an exponentially (in the number of decision variables) larger strong branching tree than selecting cuts by a simple proxy such as *efficacy* (Theorem 4.1). For branching, prior work (Dey et al., 2024) shows that strong branching can be exponentially suboptimal compared to an optimal tree.

For *perturbation instability*, we prove that an arbitrarily small perturbation of the right-hand sides in a root cut set can change the optimal B&B tree size (over all node selection and branching rules) from 1 to $2^{\Omega(n)}$, where $n$ is the number of decision variables, while changing the root LP improvement by at most $\varepsilon$ and closing nearly all of the integrality gap (Theorem 4.3). For branching, we construct instances where, for any $\varepsilon > 0$, a policy can match strong branching scores within $\varepsilon$ on all candidates at every node in either tree (one produced by strong branching or by the policy itself), yet strong branching yields at most $2n + 1$ nodes and the policy yields at least $2^{n+1} - 1$ (Theorem 5.1). Moreover, identical scores with different tie-breaking can also yield exponential gaps (Proposition 5.2). We also show that $k$ deviations from strong branching can inflate tree size by $2^{\Omega(k)}$ (Theorem 5.4). These constructions are most directly relevant to methods that train policies by imitating local expert signals such as strong branching scores or LP lookahead. They show that such imitation, while a natural and computationally attractive approach, does not guarantee global performance similar to the expert that was imitated. This gap motivates end-to-end approaches that directly optimize tree size (in a data-driven manner). For imitation learning pipelines, loss functions that account for score margins and stress tests that perturb scores or flip decisions

along the trajectory can help detect fragility.

The rest of the paper is organized as follows. Section 2 discusses related work. Section 3 formalizes the local score viewpoint and the tree size metrics we use. Sections 4 and 5 present the separations for cut selection and branching, respectively. Proofs of these results appear in the appendix. We conclude in Section 6 with implications for training and evaluation.

## 2. Related Work

**Learning score-based branch-and-cut policies.** Machine learning methods for branch-and-cut often learn a policy for selecting branching variables or cutting planes from features of the current relaxation. A common design is to train a predictor of an expensive expert signal and then apply the same $\arg\max$ rule at test time. For branching, strong branching is a standard supervision target, and many approaches aim to approximate it using learned models or parameterized scoring functions (Khalil et al., 2016; Alvarez et al., 2017; Gasse et al., 2019; Gupta et al., 2020; Zarpellon et al., 2021). For cut selection and cut management, learning objectives often use lookahead measures such as LP bound improvement or solver feedback to score candidate cuts (Paulus et al., 2022; Huang et al., 2022; Puigdemont et al., 2024; Wang et al., 2023; Tang et al., 2020). Surveys (Bengio et al., 2021; Scavuzzo et al., 2024) provide broader overviews of learning within mixed-integer optimization and branch-and-bound. A complementary line of work provides generalization guarantees for data driven algorithm design and for learning components of branch-and-bound and branch-and-cut (Gupta & Roughgarden, 2016; Balcan, 2020; Balcan et al., 2024; 2018; 2021b;a; 2022; Cheng & Basu, 2024; Cheng et al., 2024; Cheng & Basu, 2025). These results bound estimation error from finite samples, whereas our focus is on approximation and stability: whether small local imitation error or small perturbations can still lead to large changes in tree size.

**Learning beyond local score imitation.** Several approaches optimize objectives that are closer to end-to-end search effort than local score regression. One line models variable selection as a sequential decision problem and trains policies using reinforcement learning or MDP formulations, with rewards tied to downstream node counts or solve time (Etheve et al., 2020; Scavuzzo et al., 2022; Parsonson et al., 2023; Strang et al., 2025). These methods avoid strong branching scores as direct supervision, but they still face credit assignment and distribution shift along the induced search trajectory. Related work also treats branch-and-bound as a search process and trains policies from rollouts so that the learned rule is optimized for its induced trajectory rather than one step imitation at fixed

nodes (He et al., 2014). A complementary line learns primal guidance, for example diving policies, local branching, or learned large neighborhood search, to obtain good incumbents early and strengthen pruning throughout the tree (Paulus & Krause, 2023; Fischetti & Lodi, 2003; Danna et al., 2005; Liu et al., 2021; Addanki et al., 2020; Sonnerat et al., 2021; Song et al., 2020).

**Tree size and sensitivity.** The dependence of branch-and-cut performance on local choices has also been studied without learning, through lower bounds on tree size and analyses of how cutting planes interact with branching. Classical constructions show that branch-and-bound can require exponential trees even for simple integer programs (Jeroslow, 1974), and more recent work develops general lower bounds for branch-and-bound trees (Dey et al., 2023). The papers (Basu et al., 2023; 2022) study the complexity of cutting plane and branch-and-bound algorithms for mixed-integer optimization, including separations between the strength of cutting planes, branching, and their combination. From the viewpoint of cut selection, Dey and Molinaro study obstacles to selecting cutting planes using only local score information (Dey & Molinaro, 2018). Shah et al. show that strengthening the relaxation can increase tree size under a fixed branching rule, formalizing a non-monotonicity phenomenon that is relevant to cut selection (Shah et al., 2025). Finally, numerical issues and small perturbations in cut coefficients or right-hand sides are known to affect cut generation and the resulting search trajectory, motivating safe cut generation and stabilization techniques (Cook et al., 2009; Cornuéjols et al., 2013; Achterberg, 2009). Our separations are consistent with these phenomena, and they isolate two mechanisms that break the link between local supervision and global tree size.

## 3. Preliminaries

This section formalizes the local score viewpoint that motivates our analysis. We introduce notation for (i) score-based decision rules, (ii) two standard expert signals used for supervision, and (iii) the tree size metrics used in our statements.

### 3.1. Branch-and-Bound Trees

We consider mixed-integer linear programs (MILPs) of the form

$$\begin{aligned} \max \quad & \mathbf{c}^\top \mathbf{x} \\ \text{s.t.} \quad & A\mathbf{x} \le \mathbf{b}, \ \mathbf{x} \in \mathbb{Z}^{n_1} \times \mathbb{R}^{n_2}, \end{aligned}$$

where $A \in \mathbb{R}^{m \times (n_1+n_2)}$, $\mathbf{b} \in \mathbb{R}^m$, and $\mathbf{c} \in \mathbb{R}^{n_1+n_2}$, and we write $I = (A, \mathbf{b}, \mathbf{c}, n_1)$ for the resulting instance. Let $P = \{\mathbf{x} \in \mathbb{R}^{n_1+n_2} : A\mathbf{x} \le \mathbf{b}\}$ denote the LP relaxation feasible region. We say the problem is a *0-1 MILP instance*

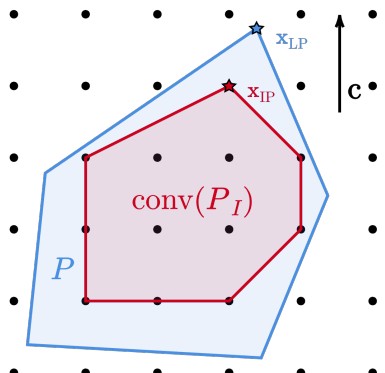

*Figure 1.* The LP relaxation $P$ (blue) contains the integer hull $\text{conv}(P_I)$ (red). Branch-and-cut adds cutting planes and branching constraints, closing the gap between the LP optimum $\mathbf{x}_{\text{LP}}$ and the integer optimum $\mathbf{x}_{\text{IP}}$.

if the integer variables are restricted to take only $0$ or $1$ values (e.g., by explicit bound constraints). The mixed-integer feasible set is $P_I = P \cap (\mathbb{Z}^{n_1} \times \mathbb{R}^{n_2})$. Figure 1 illustrates the geometric relationship between $P$, $P_I$, and their optima. A branch-and-cut (B&C) algorithm maintains a search tree whose nodes correspond to subproblems obtained from $P$ (also called the root node) by adding branching constraints and (optionally) cutting planes. At a node $N$, we write $z(N)$ for the optimal value of the node LP relaxation. The algorithm prunes nodes that are infeasible, integral, or whose LP value is at most a known incumbent value.

**Branch-and-bound.** The algorithm maintains a queue of subproblems (nodes). At each iteration, it selects a node, solves its LP relaxation, and either prunes the node (if infeasible, integral, or dominated by a known incumbent) or branches to create two child nodes that partition the feasible region. The *tree size*, defined as the number of nodes explored, is a standard measure of computational effort and the key performance metric in our analysis. To branch at a node $N$, one selects an integer variable $x_j$ that is fractional in some optimal LP solution. Branching on $x_j$ creates two children in the branch-and-bound tree by imposing $x_j \leq \lfloor x_j^* \rfloor$ and $x_j \geq \lceil x_j^* \rceil$, where $x_j^*$ is the chosen LP solution value. We write $N^{(0)}$ and $N^{(1)}$ for the two children, and

$$z^{(0)}(N, j) = z(N^{(0)}), \qquad z^{(1)}(N, j) = z(N^{(1)}).$$

The corresponding LP bound improvements are

$$\Delta^{(0)}(N, j) = z(N) - z^{(0)}(N, j),$$
$$\Delta^{(1)}(N, j) = z(N) - z^{(1)}(N, j). \qquad (1)$$

If a child is infeasible, we set its LP value to $-\infty$ and the improvement to $\infty$.

**Cutting planes.** A cutting plane is a valid inequality satisfied by all points in $P_I$ but violated by the current LP

optimum. Adding such an inequality tightens the relaxation without excluding any mixed-integer feasible point. In this paper we focus on root cuts: given a set of valid cuts $\mathcal{C}$, we add them to the root relaxation before starting branch-and-bound. This is the setting in which LP bound improvement is typically computed and used as a training signal.

### 3.2. Score-Based Decisions and Expert Signals

Many decision rules in leading solvers such as SCIP are implemented as score-based rules (Achterberg, 2009). At a node $N$, let $\mathcal{A}(N)$ denote the candidate action set, such as branching variables or candidate cuts. A *scoring rule* assigns a real value $\text{Score}(N, a)$ to each $a \in \mathcal{A}(N)$, and the induced policy selects action

$$a^* \in \arg\max_{a \in \mathcal{A}(N)} \text{Score}(N, a),$$

with ties broken by a fixed rule.

**Definition 3.1** (Score-based policy). Fix a scoring rule $\text{Score}(N, a)$. A tie-breaking rule $\tau$ maps any nonempty finite set $S$ to an element $\tau(S) \in S$. The induced policy selects, at node $N$,

$$\pi_{\text{Score}, \tau}(N) = \tau\left(\arg\max_{a \in \mathcal{A}(N)} \text{Score}(N, a)\right).$$

Machine learning approaches often train a model to predict an *expert score* from inexpensive features and then apply the above argmax rule at test time. Our results show that this local score supervision can produce much larger tree sizes compared to the expert, even when the predicted scores are uniformly close to the expert scores.

**Strong branching.** The standard expert oracle for branching is strong branching (SB). A common SB score is the product rule

$$\text{Score}_{\text{SB}}(N, j) = \max(\Delta^{(0)}(N, j), \eta_{\text{SB}})$$
$$\cdot \max(\Delta^{(1)}(N, j), \eta_{\text{SB}}), \qquad (2)$$

where a fixed constant $\eta_{\text{SB}} > 0$ ensures positivity when one improvement is zero. If either child is infeasible, we define $\text{Score}_{\text{SB}}(N, j) = \infty$. Strong branching is effective but expensive, so solvers often approximate it using proxy information such as pseudocost estimates or learned predictors (Achterberg, 2009; 2007; Khalil et al., 2016; Alvarez et al., 2017; Gasse et al., 2019; Gupta et al., 2020).

**LP bound improvement.** For a candidate cut $c \in \mathcal{A}(N)$ at the root, the expert signal used in several learning pipelines is its *LP bound improvement*: the change in the root LP value after adding the cut and resolving the LP (Coniglio & Tieves, 2015; Paulus et al., 2022; Puigdemont et al., 2024). Given an instance $I$ and a root cut

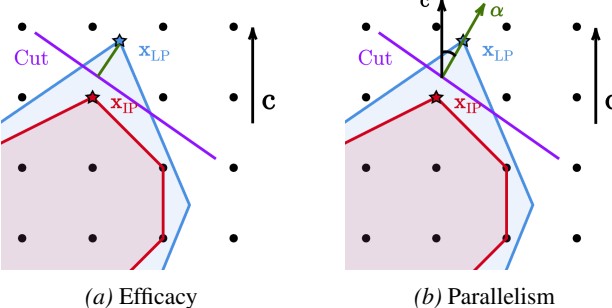

*(a) Efficacy*          *(b) Parallelism*

*Figure 2.* Proxy scores for cut selection. (a) Efficacy measures the distance from $\mathbf{x}_{\mathrm{LP}}$ to the cut. (b) Parallelism measures alignment with the objective $\mathbf{c}$.

set $\mathcal{C}$, let $z_{\mathrm{LP}}(I; \mathcal{C})$ denote the optimal value of the root LP relaxation of $I$ after adding $\mathcal{C}$ at the root, and write $z_{\mathrm{LP}}(I)$ when $\mathcal{C} = \emptyset$; the LP bound improvement is simply $z_{\mathrm{LP}}(I) - z_{\mathrm{LP}}(I; \mathcal{C})$. Evaluating this signal for many different cuts (or cut sets) requires many additional LP solves, so practical solvers and learning pipelines often also use cheaper proxy features computed from the current LP solution and the cut coefficients (Achterberg, 2009; Wesselmann & Suhl, 2012). For concreteness, for a violated cut $\boldsymbol{\alpha}^\top \mathbf{x} \leq \beta$ at the root LP optimum $\mathbf{x}_{\mathrm{LP}}$, two standard proxies are *efficacy* and *objective parallelism*:

$$\phi_{\mathrm{eff}}(\boldsymbol{\alpha}, \beta) = \frac{\boldsymbol{\alpha}^\top \mathbf{x}_{\mathrm{LP}} - \beta}{\|\boldsymbol{\alpha}\|}, \quad \phi_{\mathrm{par}}(\boldsymbol{\alpha}) = \frac{|\langle \boldsymbol{\alpha}, \mathbf{c} \rangle|}{\|\boldsymbol{\alpha}\|\|\mathbf{c}\|}.$$

Solvers use these quantities either directly as scores or as inputs to linear scoring rules. Figure 2 illustrates their geometric meaning.

### 3.3. Tree Size Metrics and Local Agreement

Our theorems compare branching policies and cut sets in terms of tree size. We use two tree size metrics, depending on whether we model a fixed solver configuration or establish a structural lower bound. Given an instance $I$, a branching policy $\pi$, and a set of root cuts $\mathcal{C}$, we write $|\mathcal{T}_\pi(I; \mathcal{C})|$ for the number of nodes produced by running branch-and-bound with best bound node selection (selecting an open node with maximum LP value, ties broken by the smallest index), branching only on variables that are fractional in the node LP optimum. When no cuts are added, we write $|\mathcal{T}_\pi(I)|$. The strong branching policy induced by (2) is denoted by $\pi_{\mathrm{SB}}$. For statements about cut selection that do not fix a branching rule, we write $|\mathcal{T}(I; \mathcal{C})|$ for the minimum number of nodes in *any* branch-and-bound tree that solves $I$ after adding $\mathcal{C}$ at the root, over all node selection and branching possibilities that branch only on LP fractional variables.

**Standing assumptions.**   Unless stated otherwise, our results hold under the following conventions. We consider

maximization problems. Node selection uses best bound, with ties broken by the smallest index. Branching is restricted to variables that are fractional in an optimal LP solution at the current node. This matches standard branch-and-bound implementations, including strong branching, and avoids branching on already integral coordinates. Our constructions respect this restriction, although allowing such branching can reduce optimal tree size on some instances (Dey et al., 2024). We count tree size as the total number of nodes, including the root. In our lower bound proofs, we may assume without loss of generality that the algorithm starts with an incumbent, meaning the objective value of a known integer feasible solution, and this value can be as large as the optimal objective value. A stronger incumbent such as this can only strengthen pruning and therefore can only decrease tree size; thus, lower bounds established under this assumption hold for any other initial incumbent, including the null incumbent. We use $N$ for B&B nodes throughout, $\Delta^{(0)}(N, j)$ and $\Delta^{(1)}(N, j)$ for the LP bound improvements in (1), and $\mathrm{Score}_{\mathrm{SB}}(N, j)$ for the strong branching score (2).

## 4. Cutting Plane Selection

This section develops the issues outlined in Section 1 for cut selection in learning pipelines.

First, the expert signal used for training can be misaligned with tree size: selecting cuts by LP bound improvement can yield exponentially larger trees than selecting by a simple proxy such as efficacy (see Section 3.2).

Second, branch-and-cut performance can be highly sensitive to small perturbations in a root cut set. Such perturbations can be amplified by the recursion, yielding exponential gaps in tree size. As a result, two root cut sets that are nearly identical under local metrics can still lead to exponentially different trees. Two mechanisms drive this instability. First, when the score gap between candidate cuts is small, a minor prediction error can flip the argmax and select a different cut. Second, repeating the same small perturbation across many blocks of a product instance can yield an exponential gap in tree size.

We present these separations in two settings. The first measures tree size under strong branching, while the second uses the minimum tree size over all possible branchings and node selections to obtain a structural lower bound.

### 4.1. Suboptimality of LP Bound Improvement

Many imitation learning approaches to cut selection use LP bound improvement as the expert signal because it measures the immediate change in the dual bound. Evaluating this signal over a large pool of candidate cuts requires many additional LP solves and can be prohibitively expensive. For

this reason, practical pipelines often rely on proxy scores such as efficacy, which can be computed from the current LP solution easily. It is natural to think that, if one could afford LP bound improvement, then one should select cuts according to it. However, the theorem below shows that this intuition can fail, even by an exponential factor.

Before stating the theorem, we note that LP bound improvement can disagree with standard proxy scores even at the root. Appendix A.1 gives a mixed-integer example with two variables that illustrates this mismatch. In that instance, LP bound improvement ranks two valid cuts in the opposite order compared to every score of the form $\lambda\phi_{\text{eff}} + (1 - \lambda)\phi_{\text{par}}$ with $\lambda \in [0, 1]$.

**Theorem 4.1** (Suboptimality of LP bound improvement). *For every $m \in \mathbb{N}$, there exists a MILP instance $I_m$ with $O(m)$ binary variables and two root cut sets $\mathcal{C}_1$ and $\mathcal{C}_2$ such that the following holds under the standing assumptions with strong branching. Consider the candidate pool $\mathcal{C}_1 \cup \mathcal{C}_2$ and a budget of $m$ cuts at the root.*

1. *Selecting the $m$ cuts with the largest LP bound improvements from $\mathcal{C}_1 \cup \mathcal{C}_2$ yields $\mathcal{C}_2$, while selecting the $m$ cuts with the largest efficacy values yields $\mathcal{C}_1$.*

2. *The resulting search trees satisfy*

$$|\mathcal{T}_{\pi_{\text{SB}}}(I_m; \mathcal{C}_1)| \le 2m + 1,$$
$$|\mathcal{T}_{\pi_{\text{SB}}}(I_m; \mathcal{C}_2)| \ge 1 + 6\left(2^{\lfloor m/9 \rfloor} - 1\right).$$

The construction (based on the two-dimensional gadget of Shah et al. (2025)) and proof appear in Appendix A.2.

Theorem 4.1 shows that, for a fixed candidate pool and cut budget, selecting root cuts by LP bound improvement or by efficacy can yield strong branching trees whose sizes differ exponentially. For learned policies, this means that even perfect prediction of LP bound improvement does not control tree size in the worst case: selecting cuts by LP bound improvement can still yield a strong branching tree that is exponentially larger than the one obtained by selecting cuts based on efficacy alone.

### 4.2. Sensitivity to Cut Perturbations

Theorem 4.1 implies that two root cut sets can yield strong branching trees whose sizes differ exponentially, even when they are selected from the same candidate pool under the same cut budget by two common cut selection rules. We further strengthen this separation by showing that such exponential gaps can arise even when the two cut sets are nearly identical, differing only by an arbitrarily small perturbation of their right-hand sides.

Our construction builds a product instance from $m$ disjoint copies of a base gadget. This is in the spirit of Basu et al.

(2023, Theorem 2.2), which gives a $2^{m+1} - 1$ lower bound on the B&B tree size for the stable set problem on $m$ disjoint triangles. The lemma below abstracts the property behind this product lower bound and extends it to more general problems: for each coordinate, there are optimal integer solutions that take different values on that coordinate.

**Lemma 4.2.** *Let $d \in \mathbb{N}$, let $B \subseteq [0, 1]^d$ be a nonempty compact convex set, and let $\mathbf{w} \in \mathbb{R}^d$. Define $\beta = \max\left\{\langle \mathbf{w}, \mathbf{x}\rangle : \mathbf{x} \in B \cap \{0, 1\}^d\right\}$. Assume:*

*(i)* $\max\left\{\langle \mathbf{w}, \mathbf{x}\rangle : \mathbf{x} \in B\right\} > \beta$, *and*

*(ii) for every $i \in [d]$, there exist $\mathbf{u}, \mathbf{v} \in B \cap \{0, 1\}^d$ with $\langle \mathbf{w}, \mathbf{u}\rangle = \langle \mathbf{w}, \mathbf{v}\rangle = \beta$ and $u_i \ne v_i$.*

*For $m \in \mathbb{N}$, consider the 0-1 MILP*

$$\max\left\{\sum_{t=1}^{m} \langle \mathbf{w}, \mathbf{x}^{(t)}\rangle : \mathbf{x} \in B^m \cap \{0, 1\}^{dm}\right\},$$

*where $\mathbf{x} = (\mathbf{x}^{(1)}, \ldots, \mathbf{x}^{(m)}) \in \underbrace{\mathbb{R}^d \times \ldots \times \mathbb{R}^d}_{m \text{ times}} = \mathbb{R}^{dm}$, each $\mathbf{x}^{(t)} = (x_1^{(t)}, \ldots, x_d^{(t)})$, and $B^m = \underbrace{B \times \ldots \times B}_{m \text{ times}}$. Then the mixed-integer optimum equals $m\beta$, and every branch-and-bound tree solving this MILP using only variable disjunctions, i.e., disjunctions of the form $x_i^{(t)} \le 0$ or $x_i^{(t)} \ge 1$ for some $(t, i) \in [m] \times [d]$, has at least $2^{m+1} - 1$ nodes.*

The proof is an induction on $m$: the condition on coordinates ensures that both children of any root branching decision contain optimal solutions, preventing pruning. The full proof appears in Appendix A.3.

We now state our main result on cut perturbations. We apply Lemma 4.2 to the stable set problem on $m$ disjoint triangles and compare two root cut sets whose right-hand sides differ by an arbitrarily small amount.

**Theorem 4.3** (Exponential gap from tiny cut perturbations). *For any $\varepsilon > 0$ and any integer $n \ge 4$, let $m = \lfloor n/3 \rfloor$ and $\varepsilon' = \min\left\{\frac{1}{4}, \frac{\varepsilon}{m}\right\}$. There exists a mixed-integer linear program instance $I$ with $n$ variables and two root cut sets of valid cutting planes, $\mathcal{C}$ and $\widetilde{\mathcal{C}}$, such that the following hold.*

1. *The cut sets $\mathcal{C}$ and $\widetilde{\mathcal{C}}$ can be paired so that each pair has identical coefficients and their right-hand sides differ by $\varepsilon'$.*

2. *The root LP values satisfy*

$$|z_{\text{LP}}(I; \mathcal{C}) - z_{\text{LP}}(I; \widetilde{\mathcal{C}})| \le m\varepsilon' \le \varepsilon.$$

3. *Both cut sets close at least a $(1 - 2\varepsilon')$ fraction of the root LP gap.*

*4. The minimum tree sizes satisfy*

$$|\mathcal{T}(I;\mathcal{C})| = 1 \quad and \quad |\mathcal{T}(I;\widetilde{\mathcal{C}})| = 2^{m+1} - 1.$$

The construction and proof appear in Appendix A.3.

At a high level, the construction starts from the stable set problem on $m$ disjoint triangles, as in Basu et al. (2023, Theorem 2.2). We compare the cut set that enforces $x_1^{(t)} + x_2^{(t)} + x_3^{(t)} \le 1$ in every block with a weakened version whose right-hand side is $1 + \varepsilon'$. Both cut sets close almost all of the root LP gap, but the weaker cut set yields an exponential minimum tree size.

*Remark* 4.4. In the construction of Theorem 4.3, the minimum tree size changes from 1 to $2^{m+1} - 1$ under a perturbation that is arbitrarily small under common proxy scores evaluated at the root. Consider a paired cut $c : \boldsymbol{\alpha}^\top \mathbf{x} \le \beta$ in $\mathcal{C}$ and $\tilde{c} : \boldsymbol{\alpha}^\top \mathbf{x} \le \beta + \varepsilon'$ in $\widetilde{\mathcal{C}}$, and let $\mathbf{x}_{\mathrm{LP}}$ be the root LP optimum of $I$ before adding any cuts. Then $\phi_{\mathrm{par}}(\boldsymbol{\alpha})$ is identical for $c$ and $\tilde{c}$, and the efficacies satisfy

$$\left| \phi_{\mathrm{eff}}(\boldsymbol{\alpha}, \beta) - \phi_{\mathrm{eff}}(\boldsymbol{\alpha}, \beta + \varepsilon') \right| = \frac{\varepsilon'}{\|\boldsymbol{\alpha}\|}.$$

In particular, for any $\lambda \in [0, 1]$, the proxy score $\lambda \phi_{\mathrm{eff}} + (1 - \lambda)\phi_{\mathrm{par}}$ differs by at most $\lambda \varepsilon'/\|\boldsymbol{\alpha}\|$ across the pair. Thus, even a small estimation error in a learned proxy score can reverse the ranking between $c$ and $\tilde{c}$ when the proxy score gap is small.

*Remark* 4.5. Theorem 4.3 also reveals two general insights relevant to the broader integer programming community:

1. The B&C tree size can be highly sensitive to small changes in cut definitions. Theorem 4.3 shows that a small perturbation of the right-hand sides in a root cut set can lead to exponentially different tree sizes. This matters because slightly changing an inequality's right-hand side is a common technique used to ensure cut validity in the presence of numerical rounding errors (Cook et al., 2009; Cornuéjols et al., 2013).

2. Closing a large fraction of the integrality gap does not necessarily imply a small tree size. In the construction, the cut set $\widetilde{\mathcal{C}}$ closes $(1 - 2\varepsilon')$ of the gap, which can be arbitrarily close to 1, yet branch-and-bound still requires exponentially many nodes. This shows that gap closure alone does not control tree size in the worst case, although empirical studies of full strong branching suggest that tree size often decreases once a set of cuts closes a substantial fraction of the integrality gap (Shah et al., 2025).

## 5. Branching Variable Selection

For branching variable selection, the same issues from Section 1 arise. First, the expert oracle can be misaligned

with tree size, since strong branching can be exponentially suboptimal (Dey et al., 2024). Second, branch-and-bound recursion can amplify small errors in local scores, or a small number of deviations from an expert, leading to exponential gaps in tree size. This section focuses on the second obstacle and establishes two separations. First, for an explicit family $\{\widetilde{I}_n\}$ of MILP instances, for every $\varepsilon > 0$ we construct a score-based policy $\hat{\pi}$ whose scoring function satisfies $|\widehat{\mathrm{Score}} - \mathrm{Score}_{\mathrm{SB}}| \le \varepsilon$ on every candidate variable at every node visited by either $\pi_{\mathrm{SB}}$ or $\hat{\pi}$, yet the resulting tree size from $\hat{\pi}$ is exponentially larger than the tree size from $\pi_{\mathrm{SB}}$ (Theorem 5.1). Second, we show that $k$ deviations from strong branching along its trajectory can already inflate tree size by $2^{\Omega(k)}$ (Theorem 5.4).

Our constructions rely on two effects. When the score gap between candidate variables is small, a minor prediction error can change the argmax and alter the branching choice. Moreover, a single incorrect branching decision can move the search to a different part of the tree, after which subsequent decisions can compound the deviation. Together, these effects can turn small local differences into exponential gaps in tree size.

### 5.1. Exponential Gap from Small Score Differences

We now present our main theoretical result on branching, which shows that arbitrarily small uniform score discrepancies can lead to exponentially different trees.

**Theorem 5.1** (Exponential gap from arbitrarily small score differences)**.** *There exists a family of $0$–$1$ MILP instances $\{\widetilde{I}_n\}_{n\in\mathbb{N}}$, where each $\widetilde{I}_n$ has $O(n)$ variables, such that the following holds under the standing assumptions. For every $n \ge 3$ and every $\varepsilon > 0$, there exists a branching policy $\hat{\pi}$ with scoring function $\widehat{\mathrm{Score}}$ satisfying:*

1. *For every node $N$ in the B&B tree generated by either $\pi_{\mathrm{SB}}$ or $\hat{\pi}$ on $\widetilde{I}_n$, and every candidate variable index $j$,*

$$\left| \widehat{\mathrm{Score}}(N, j) - \mathrm{Score}_{\mathrm{SB}}(N, j) \right| \le \varepsilon.$$

2. *The tree sizes diverge exponentially:*

$$|\mathcal{T}_{\pi_{\mathrm{SB}}}(\widetilde{I}_n)| \le 2n + 1 \quad and \quad |\mathcal{T}_{\hat{\pi}}(\widetilde{I}_n)| \ge 2^{n+1} - 1.$$

The construction and proof appear in Appendix B.2.

Theorem 5.1 reveals a fundamental instability of score-based branching policies. Even when a scoring function matches strong branching scores up to an arbitrarily small uniform error, the resulting trees can differ exponentially in size. This shows that small score differences, whether from learning error or numerical perturbations, can lead to large changes in solver performance. The exponential gap arises because a small score perturbation can change

the set of maximizers, and under a fixed tie-breaking rule this can change the selected branching variable at many nodes. These local changes can then compound through the recursive structure of branch-and-bound.

A score-based policy consists of a scoring function and a tie-breaking rule as in Definition 3.1. Theorem 5.1 shows that under a fixed tie-breaking rule, arbitrarily small score discrepancies can lead to an exponential gap in tree size. The next proposition shows that even when the scoring function is held fixed, changing only the tie-breaking rule can still lead to an exponential gap.

**Proposition 5.2** (Exponential gap from tie-breaking under identical scores)**.** *There exists a family of 0–1 MILP instances* $\{I_n\}_{n \geq 3}$, *where each* $I_n$ *has* $O(n)$ *variables, and a scoring function* $\mathrm{Score}(N, j)$ *such that the following holds under the standing assumptions. There exist two score-based branching policies* $\pi_{\min}$ *and* $\pi_y$ *that use the same scoring function* $\mathrm{Score}$ *and differ only in their tie-breaking rule among maximizers. For every* $n \geq 3$,

$$|\mathcal{T}_{\pi_{\min}}(I_n)| = 2n + 1 \qquad and \qquad |\mathcal{T}_{\pi_y}(I_n)| \geq 2^{n+1} - 1.$$

The construction and proof appear in Appendix B.3.

### 5.2. Sensitivity to Sparse Deviations

While Theorem 5.1 shows that small, persistent differences in scoring can lead to exponential gaps, it is also important to understand the impact of a small number of decision differences from a baseline policy. Ye et al. (2023) mention the sensitivity of B&B to branching decisions, remarking that "one wrong decision may cause a doubled tree size." In imitation learning, a standard way to measure deviations is along the trajectory induced by an expert: one runs strong branching and counts how often the learned policy makes a different branching choice on the internal nodes of that run. The next theorem shows that even this restricted notion of deviation does not control the resulting tree size: $k$ deviations can already lead to exponential (in $k$) blowup in size. To state this precisely, we first define what it means for a policy to differ from strong branching along its trajectory.

**Definition 5.3** (Deviations along the strong branching run)**.** Fix an instance $I$. For any branching policy $\pi$, let $\mathcal{U}_\pi(I)$ denote the set of internal nodes of $\mathcal{T}_\pi(I)$. For any B&B node $N$, write $\pi(N)$ for the branching variable selected by $\pi$ at $N$. We say that $\pi$ differs from $\pi_{\mathrm{SB}}$ on exactly $k$ branching decisions along the strong branching run on $I$ if

$$\left|\{N \in \mathcal{U}_{\pi_{\mathrm{SB}}}(I) : \pi(N) \neq \pi_{\mathrm{SB}}(N)\}\right| = k.$$

**Theorem 5.4** (Exponential growth from $k$ deviations)**.** *For every* $n \in \mathbb{N}$, *there exists a packing instance* $I_n$ *such that for every* $k \in \{0, 1, \ldots, n\}$ *there exists a branching policy* $\hat{\pi}_{n,k}$ *satisfying the following under the standing assumptions.*

1. *Policy* $\hat{\pi}_{n,k}$ *differs from* $\pi_{\mathrm{SB}}$ *on exactly* $k$ *branching decisions along the strong branching run on* $I_n$.

2. *The resulting tree sizes satisfy*

$$|\mathcal{T}_{\hat{\pi}_{n,k}}(I_n)| \geq 2^{\Omega(k)}|\mathcal{T}_{\pi_{\mathrm{SB}}}(I_n)|.$$

The construction and proof appear in Appendix B.

Definition 5.3 counts deviations on the internal nodes $\mathcal{U}_{\pi_{\mathrm{SB}}}(I)$ visited by the strong branching rollout. This matches the standard imitation learning setting where training data and offline evaluation are collected on expert trajectories. From a solver perspective, one may instead count deviations on the internal nodes visited by $\hat{\pi}_{n,k}$. Define

$$k' := \left|\{N \in \mathcal{U}_{\hat{\pi}_{n,k}}(I_n) : \hat{\pi}_{n,k}(N) \neq \pi_{\mathrm{SB}}(N)\}\right|.$$

For the instance $I_n$ and policy $\hat{\pi}_{n,k}$ in Theorem 5.4, the proof in Appendix B implies

$$|\mathcal{T}_{\hat{\pi}_{n,k}}(I_n)| \geq \Omega(k') \, |\mathcal{T}_{\pi_{\mathrm{SB}}}(I_n)|.$$

## 6. Conclusion and Discussion

We study the inherent limitations in the use of local characteristics of a B&C tree, such as strong branching and LP objective improvement, to guide branching and cut selection decisions. We identify two main pitfalls. First, such local "scores" may result in exponentially suboptimal decisions, even if they are computed exactly (already observed for strong branching in earlier work by (Dey et al., 2024)). Second, if one approximates these "scores" using methods based on learning or other approaches for computational efficiency, then even arbitrarily small, but nonzero, approximation errors can lead to exponential blowups in tree sizes, even when the original "score" produces small B&C trees.

For cut selection, we showed that LP bound improvement can be a bad indicator of overall tree size by exhibiting instances where selecting cuts by LP bound improvement yields an exponentially larger strong branching tree than selecting cuts by a proxy score, and we proved that an arbitrarily small perturbation of the right-hand sides in a root cut set can change the optimal B&B tree size from 1 to $2^{\Omega(n)}$ while barely affecting the root LP improvement. For branching, we showed that arbitrarily small uniform deviations from strong branching scores can produce exponentially larger trees, that identical scores with different tie-breaking can also yield exponential gaps, and that $k$ deviations from strong branching can induce a $2^{\Omega(k)}$ increase in tree size.

Our lower bounds are worst case statements and do not suggest that strong branching, LP bound improvement, or their learned approximations should be discarded in practice. Instead, they highlight two aspects that are invisible to standard local evaluation, namely the quality of the expert

itself and the stability of the induced search under small perturbations. In empirical work, this suggests reporting not only average tree sizes but also how performance changes when predicted scores are perturbed, ties are broken differently, or a small fraction of branching decisions is forced to follow an alternative rule. Such stress tests can help assess whether a learned policy is fragile on its target distribution.

These results complement existing generalization analyses by isolating an approximation barrier in data-driven methods based on learning: local accuracy on expert signals does not guarantee good global performance. In other words, imitating an expert is a natural proxy for learning a good policy, but it can fail to control tree size in the worst case. This gap motivates end-to-end training and learning that directly optimizes global objectives such as tree size. Reinforcement learning with a reward based on tree size is one such approach; it avoids the mismatch between training signal and evaluation metric. For imitation learning pipelines that remain attractive due to their computational or sample efficiency, losses that account for score margins and stress tests (e.g., perturbing predicted scores, randomizing tie-breaking, or flipping decisions along the trajectory) can help detect and mitigate fragility. Identifying conditions under which local supervision provably controls tree size, and characterizing instance classes where such conditions hold, remain open directions.

## Impact Statement

This paper presents work whose goal is to advance the field of machine learning. There are many potential societal consequences of our work, none of which we feel must be specifically highlighted here.

## Acknowledgements

Hongyu Cheng and Amitabh Basu gratefully acknowledge support from the Air Force Office of Scientific Research (AFOSR) grant FA9550-25-1-0038. Hongyu Cheng also acknowledges support from the Johns Hopkins University Mathematical Institute for Data Science (MINDS) Fellowship and the Duncan Award.

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

# A. Proofs and Examples for Cutting Plane Selection

## A.1. A Mixed-Integer Example with Two Variables

We give an example in which LP bound improvement and standard proxy scores rank two valid cuts in opposite orders, even at the root. Consider the mixed-integer program

$$\max \ -y$$
$$\text{s.t. } x + y = s$$
$$x \in \mathbb{Z}_+, \ y \in \mathbb{R}_+,$$

where $s \in (1, 2)$. The LP relaxation (dropping integrality of $x$) has the unique optimum $(x, y) = (s, 0)$ with value 0, since the objective maximizes $-y$ subject to $y \geq 0$. Moreover, $y \geq 0$ implies $x \leq s < 2$, so the mixed-integer feasible set consists of the two points $(0, s)$ and $(1, s - 1)$.

Fix $a, b > 0$ and consider an inequality of the form

$$-ax - by \leq -1.$$

Requiring validity for the two mixed-integer points yields

$$bs \geq 1 \qquad \text{and} \qquad a + b(s - 1) \geq 1,$$

while requiring violation at the root LP optimum $(s, 0)$ yields $as < 1$. In particular, $bs \geq 1$ and $as < 1$ imply $b > a$. For such a cut, the efficacy and objective parallelism at the root are

$$\phi_{\text{eff}}(s, a, b) = \frac{1 - sa}{\sqrt{a^2 + b^2}}, \qquad \phi_{\text{par}}(a, b) = \frac{b}{\sqrt{a^2 + b^2}}.$$

To compute the LP bound improvement, eliminate $y$ using $y = s - x$. The cut becomes

$$-ax - b(s - x) \leq -1 \qquad \Longleftrightarrow \qquad (b - a)x \leq bs - 1.$$

Since $b > a$, this is equivalent to a lower bound on $y$,

$$y = s - x \geq \Delta(s, a, b), \qquad \Delta(s, a, b) = \frac{1 - sa}{b - a}.$$

Thus the root LP value decreases from 0 to $-\Delta(s, a, b)$, and the LP bound improvement equals $\Delta(s, a, b)$.

We now fix $s = 3/2$ and compare two valid cuts:

$$\text{(cut 1)} \quad -\tfrac{1}{2}x - y \leq -1, \qquad \text{(cut 2)} \quad -\tfrac{1}{3}x - 2y \leq -1.$$

Both inequalities are satisfied by $(0, 3/2)$ and $(1, 1/2)$ and are violated by the root LP optimum $(3/2, 0)$. A direct calculation gives $\Delta_1 = \tfrac{1}{2}$ and $\Delta_2 = \tfrac{3}{10}$, so LP bound improvement ranks cut 1 above cut 2. In contrast, cut 2 has larger values for both proxy scores:

$$\phi_{\text{eff},1} = \frac{1}{2\sqrt{5}} < \frac{3}{2\sqrt{37}} = \phi_{\text{eff},2}, \qquad \phi_{\text{par},1} = \frac{2}{\sqrt{5}} < \frac{6}{\sqrt{37}} = \phi_{\text{par},2}.$$

Therefore, for any $\lambda \in [0, 1]$, the proxy score $\lambda\phi_{\text{eff}} + (1 - \lambda)\phi_{\text{par}}$ ranks cut 2 above cut 1, while LP bound improvement ranks cut 1 above cut 2.

## A.2. Proof of Theorem 4.1

This example is based on the two-dimensional gadget from Shah et al. (2025).

*Proof of Theorem 4.1.* Let

$$P = \left\{ (x, y) \in [0, 1]^2 : -7x + y \leq 0.3, \ 5x + 8y \leq 8.5, \ 3x + 2y \leq 3.7 \right\}$$

and let $\mathbf{c} = (6, 5) \in \mathbb{R}^2$. For $m \in \mathbb{N}$, consider the 0–1 MILP $I_m$ with binary variables $(x_i, y_i) \in \{0, 1\}^2$ for $i \in [m]$:

$$\max \quad \sum_{i=1}^{m} (6x_i + 5y_i)$$
$$\text{s.t.} \quad (x_i, y_i) \in P \qquad i \in [m].$$

We refer to the pair $(x_i, y_i)$ and the constraint $(x_i, y_i) \in P$ as block $i$. Define the root cut sets

$$\mathcal{C}_1 = \{20y_i - 7x_i \leq 0 : i \in [m]\}, \qquad \mathcal{C}_2 = \{13x_i + 10y_i \leq 14 : i \in [m]\}.$$

A direct check shows that $P \cap \{0, 1\}^2 = \{(0, 0), (1, 0)\}$, and both inequalities are satisfied by these two points. Hence every cut in $\mathcal{C}_1 \cup \mathcal{C}_2$ is valid for $I_m$. Moreover, the root LP optimum in each block is $A = (0.9, 0.5)$, and it violates both inequalities, so both cut sets are cutting planes at the root. We begin with the polytope $P$ for a single block in variables $(x, y)$. A direct enumeration of feasible intersections of the defining inequalities shows that $P$ has vertices

$$F = (0, 0), \ E = (0, 0.3), \ D = (0.1, 1), \ C = (1, 0), \ B = (1, 0.35), \ A = (0.9, 0.5).$$

Figure 3 depicts $P$ together with the two blockwise cuts from $\mathcal{C}_1$ and $\mathcal{C}_2$. Their objective values under $\mathbf{c} = (6, 5)$ are

$$\mathbf{c}^\top A = \frac{79}{10}, \quad \mathbf{c}^\top B = \frac{31}{4}, \quad \mathbf{c}^\top C = 6, \quad \mathbf{c}^\top D = \frac{28}{5}, \quad \mathbf{c}^\top E = \frac{3}{2}, \quad \mathbf{c}^\top F = 0,$$

so the LP relaxation of a single block is uniquely optimized at $A$.

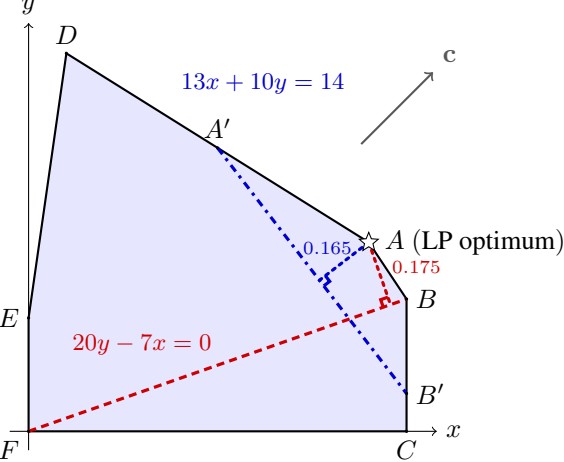

*Figure 3.* The polytope $P$ for one block and the two cuts induced by $\mathcal{C}_1$ and $\mathcal{C}_2$. The dotted segments from $A$ to the cut lines are perpendicular, and their lengths equal the corresponding efficacies at $A$.

We first compute the root LP value. In the root LP relaxation of $I_m$, each block can attain $A$, hence $z_{\mathrm{LP}}(I_m) = m \cdot \frac{79}{10}$.

We now compare LP bound improvements for individual cuts at the root. Fix any $i \in [m]$ and let $c_{2,i}$ denote the cut $13x_i + 10y_i \leq 14$ (shown in blue in Figure 3) and $c_{1,i}$ denote the cut $20y_i - 7x_i \leq 0$ (shown in red in Figure 3). Since the instance decomposes across blocks, adding one cut affects only the corresponding block.

For $c_{2,i}$, the affected block has feasible region $P \cap \{13x + 10y \leq 14\}$ and a unique optimum at

$$A' = (0.5, 0.75), \qquad \mathbf{c}^\top A' = \frac{27}{4},$$

since $A'$ is the intersection of $13x + 10y = 14$ and $5x + 8y = 8.5$, and $\frac{27}{4}$ dominates the objective values at the other vertices of this intersection, including $B' = (1, 0.1)$ with value $\frac{13}{2}$. Thus $z_{\mathrm{LP}}(I_m; \{c_{2,i}\}) = (m-1) \cdot \frac{79}{10} + \frac{27}{4}$ and the LP bound improvement equals $\frac{79}{10} - \frac{27}{4} = \frac{23}{20}$.

For $c_{1,i}$, on $[0,1]^2$ the inequality $20y - 7x \leq 0$ is equivalent to $y \leq 0.35x$, and together with $3x + 2y \leq 3.7$ it yields the triangle $\operatorname{conv}\{F, C, B\}$. Since $\mathbf{c}^\top B = \frac{31}{4}$ dominates $\mathbf{c}^\top C$ and $\mathbf{c}^\top F$, the unique block optimum is $B$. Thus $z_{\mathrm{LP}}(I_m; \{c_{1,i}\}) = (m-1) \cdot \frac{79}{10} + \frac{31}{4}$ and the LP bound improvement equals $\frac{79}{10} - \frac{31}{4} = \frac{3}{20}$.

Therefore every cut in $\mathcal{C}_2$ has strictly larger LP bound improvement than every cut in $\mathcal{C}_1$, and selecting the $m$ cuts with the largest LP bound improvements returns $\mathcal{C}_2$. Adding all $m$ cuts from $\mathcal{C}_2$ yields $z_{\mathrm{LP}}(I_m; \mathcal{C}_2) = m \cdot \frac{27}{4}$ and $z_{\mathrm{LP}}(I_m) - z_{\mathrm{LP}}(I_m; \mathcal{C}_2) = \frac{23m}{20}$. Adding all $m$ cuts from $\mathcal{C}_1$ yields $z_{\mathrm{LP}}(I_m; \mathcal{C}_1) = m \cdot \frac{31}{4}$ and $z_{\mathrm{LP}}(I_m) - z_{\mathrm{LP}}(I_m; \mathcal{C}_1) = \frac{3m}{20}$.

For efficacy, we use $\phi_{\mathrm{eff}}$ from the main text and evaluate it at the root LP optimum $A = (0.9, 0.5)$ of a free block. For any cut $c \in \mathcal{C}_2$, the violation at $A$ is $13 \cdot 0.9 + 10 \cdot 0.5 - 14 = \frac{27}{10}$ and the normal has Euclidean norm $\sqrt{13^2 + 10^2} = \sqrt{269}$, so $\phi_{\mathrm{eff}}(c) = \frac{27}{10\sqrt{269}}$. For any cut $c \in \mathcal{C}_1$, the violation at $A$ is $20 \cdot 0.5 - 7 \cdot 0.9 = \frac{37}{10}$ and the normal has Euclidean norm $\sqrt{(-7)^2 + 20^2} = \sqrt{449}$, so $\phi_{\mathrm{eff}}(c) = \frac{37}{10\sqrt{449}}$. Since $\frac{37}{10\sqrt{449}} > \frac{27}{10\sqrt{269}}$, every cut in $\mathcal{C}_1$ has strictly larger efficacy than every cut in $\mathcal{C}_2$, and selecting the $m$ cuts with the largest efficacy values returns $\mathcal{C}_1$.

We now analyze the tree size after adding $\mathcal{C}_1$. Under $\mathcal{C}_1$, each unfixed block has unique LP optimum $B = (1, 0.35)$, so $x_i$ is integral and $y_i$ is fractional. Strong branching considers only fractional binary variables, so it branches on some $y_i$ from an unfixed block. The child $y_i = 1$ is infeasible since $y_i \leq 0.35x_i \leq 0.35$. The feasible child $y_i = 0$ has block optimum $C = (1, 0)$, which is integral. Thus each branching fixes one additional block integrally and creates one infeasible sibling. The search tree is a chain of $m$ branchings with $m$ infeasible siblings, so $|\mathcal{T}_{\pi_{\mathrm{SB}}}(I_m; \mathcal{C}_1)| \leq 2m + 1$.

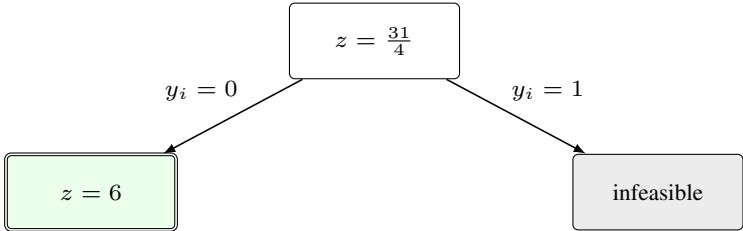

*Figure 4.* The gadget under $\mathcal{C}_1$ in one block. Here $z$ denotes the block LP value. Branching on $y_i$ yields one integral child and one infeasible child.

We now analyze the tree size after adding $\mathcal{C}_2$. We follow the argument in the proof of Theorem 2 in Shah et al. (2025) and include the main steps here so that the paper is self contained.

We use the strong branching notation from Section 3. At a node $N$, let $z(N)$ denote its LP value. For an index $j$ that is fractional in an optimal LP solution at $N$, let $z^{(0)}(N, j)$ and $z^{(1)}(N, j)$ denote the child LP values obtained by fixing $x_j = 0$ and $x_j = 1$. Recall $\Delta^{(0)}(N, j) = z(N) - z^{(0)}(N, j)$ and $\Delta^{(1)}(N, j) = z(N) - z^{(1)}(N, j)$, and the strong branching product score

$$\mathrm{Score}_{\mathrm{SB}}(N, j) = \max\left\{\Delta^{(0)}(N, j), \eta_{\mathrm{SB}}\right\} \cdot \max\left\{\Delta^{(1)}(N, j), \eta_{\mathrm{SB}}\right\}.$$

Consider an unfixed block $i$. Under $\mathcal{C}_2$, this block has unique LP optimum $A' = (0.5, 0.75)$, so both $x_i$ and $y_i$ are fractional. We compare strong branching scores for branching on $x_i$ and $y_i$ using (2). We break ties by the smallest index, and we index variables so that $x_i$ precedes $y_i$ within each block.

If we branch on $x_i$, then in the child $x_i = 0$ the block optimum is $E = (0, 0.3)$ with value $\frac{3}{2}$, and in the child $x_i = 1$ the block optimum is $B' = (1, 0.1)$ with value $\frac{13}{2}$. The two improvements are $\frac{27}{4} - \frac{3}{2} = \frac{21}{4}$ and $\frac{27}{4} - \frac{13}{2} = \frac{1}{4}$, so the score in (2) equals

$$\max\left(\frac{21}{4}, \eta_{\mathrm{SB}}\right) \cdot \max\left(\frac{1}{4}, \eta_{\mathrm{SB}}\right).$$

If we branch on $y_i$, then in the child $y_i = 0$ the block optimum is $C = (1, 0)$ with value 6, and in the child $y_i = 1$ the block optimum is $D = (0.1, 1)$ with value $\frac{28}{5}$. The two improvements are $\frac{27}{4} - 6 = \frac{3}{4}$ and $\frac{27}{4} - \frac{28}{5} = \frac{23}{20}$, so the score in (2) equals

$$\max\left(\frac{3}{4}, \eta_{\mathrm{SB}}\right) \cdot \max\left(\frac{23}{20}, \eta_{\mathrm{SB}}\right).$$

If $\eta_{\mathrm{SB}} < \frac{3}{4}$, then the score for $y_i$ equals $\frac{3}{4} \cdot \frac{23}{20} = \frac{69}{80}$, while the score for $x_i$ is at least $\frac{21}{4} \cdot \frac{1}{4} = \frac{21}{16}$. If $\frac{3}{4} \leq \eta_{\mathrm{SB}} < \frac{21}{4}$, then the score for $x_i$ equals $\frac{21}{4} \eta_{\mathrm{SB}}$, while the score for $y_i$ equals $\eta_{\mathrm{SB}} \max\left(\frac{23}{20}, \eta_{\mathrm{SB}}\right) < \frac{21}{4} \eta_{\mathrm{SB}}$. Therefore the score for $x_i$ is strictly larger whenever $\eta_{\mathrm{SB}} < \frac{21}{4}$. When $\eta_{\mathrm{SB}} \geq \frac{21}{4}$, the two scores tie and both equal $\eta_{\mathrm{SB}}^2$, and the smallest index among the maximizers corresponds to an $x_i$ from an unfixed block. Therefore, in any node with at least one unfixed block, strong branching selects some $x_i$ from an unfixed block. In either child $x_i = 0$ or $x_i = 1$, the LP optimum has $y_i$ fractional and the branch $y_i = 1$ is infeasible, since $y_i \leq 0.3$ when $x_i = 0$ and $y_i \leq 0.1$ when $x_i = 1$. Thus the strong branching score of $y_i$ is infinite in either child, and the next branching is on $y_i$. After branching on $x_i$ and then on $y_i$ in block $i$, exactly two grandchildren are feasible, namely $(x_i, y_i) = (0, 0)$ and $(x_i, y_i) = (1, 0)$, and both are integral for that block.

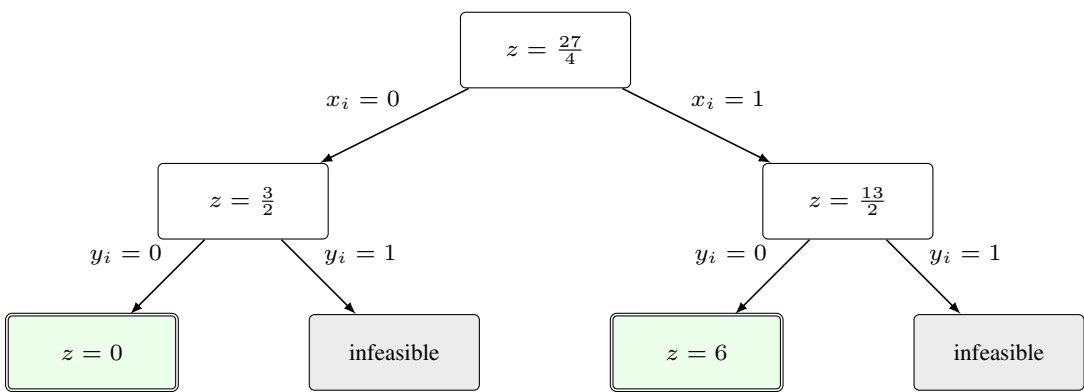

*Figure 5.* The gadget under $\mathcal{C}_2$ in one block. Here $z$ denotes the block LP value. Strong branching branches on $x_i$ and then on $y_i$, and the $y_i = 1$ children are infeasible.

Fix $k = \lfloor m/9 \rfloor$. Let $N$ be any node before $k$ blocks have been fixed integrally in this manner. Let $d$ be the number of blocks that have already been fixed integrally along the path to $N$. If $N$ is at even depth with $d$ completed blocks, then at least $m - d$ blocks are still unfixed, each contributing $\frac{27}{4}$ to the LP value, and fixed blocks contribute at least 0. Hence $z(N) \geq \frac{27}{4}(m - d)$. If $N$ is at odd depth, then one additional block has $x$ fixed but $y$ unfixed. In that partially fixed block, the LP value is at least $\frac{3}{2}$, so

$$z(N) \geq \frac{3}{2} + \frac{27}{4}(m - d - 1) = \frac{27}{4}(m - d) - \frac{21}{4}.$$

Since $P \cap \{0,1\}^2 = \{(0,0), (1,0)\}$, the integer optimum of $I_m$ equals $6m$. Thus any node $N$ with $z(N) > 6m$ cannot be pruned by bound, even if an incumbent of value $6m$ is available. For $d \leq k - 1$ we have $m - 9d \geq m - 9(k - 1) \geq 9$, so

$$\frac{27}{4}(m - d) - 6m = \frac{3}{4}(m - 9d) > 0$$

and

$$\frac{27}{4}(m - d) - \frac{21}{4} - 6m = \frac{3}{4}(m - 9d - 7) > 0.$$

Therefore no node created before completing $k$ blocks can be pruned by bound. Such a node is also not pruned by integrality, since it still contains an unfixed block.

This yields a lower bound on the size of the resulting tree that does not depend on the order in which open nodes are processed. For $t \in \{0, 1, \ldots, k\}$, after completing $t$ blocks, the tree contains $2^t$ feasible nodes at depth $2t$. For $t < k$, none of these nodes can be pruned by bound or integrality, so each must eventually be branched on, completing one additional block and producing two feasible grandchildren. Each such completion adds exactly 6 new nodes, namely two $x$ children and their four $y$ children. Summing over $t = 0, \ldots, k - 1$ gives that the total number of nodes is at least

$$1 + \sum_{t=0}^{k-1} 6 \cdot 2^t = 1 + 6\left(2^k - 1\right) = 1 + 6\left(2^{\lfloor m/9 \rfloor} - 1\right),$$

which yields the stated lower bound. $\qquad \square$

## A.3. Proof of Theorem 4.3

We prove Theorem 4.3 via a product lower bound for branch-and-bound trees.

*of Lemma 4.2.* The integer optimum equals $m\beta$. Indeed, for $\mathbf{x} \in B^m \cap \{0,1\}^{dm}$ each block $\mathbf{x}^{(t)}$ belongs to $B \cap \{0,1\}^d$, hence $\langle \mathbf{w}, \mathbf{x}^{(t)} \rangle \leq \beta$, and summing over $t \in [m]$ gives $\sum_{t=1}^m \langle \mathbf{w}, \mathbf{x}^{(t)} \rangle \leq m\beta$. Conversely, if $\mathbf{x}^\star \in B \cap \{0,1\}^d$ attains $\beta$, then $(\mathbf{x}^\star, \dots, \mathbf{x}^\star) \in B^m \cap \{0,1\}^{dm}$ attains $m\beta$.

We now prove the tree size lower bound. We claim that for every $m$, every branch-and-bound tree that solves the instance with $m$ blocks using only disjunctions of the form $x_i^{(t)} \leq 0$ or $x_i^{(t)} \geq 1$ for some $t \in [m]$ and $i \in [d]$ has at least $2^{m+1} - 1$ nodes. Since providing a stronger incumbent can only decrease the number of explored nodes, we may assume the algorithm is warm started with an incumbent of value $m\beta$. Thus every leaf is either infeasible or has LP upper bound at most $m\beta$. We prove the claim by induction on $m$.

**Base case.** Fix $m = 1$ and let $\mathcal{T}$ be any such tree. By assumption (i), the root LP value satisfies $\max\{\langle \mathbf{w}, \mathbf{x} \rangle : \mathbf{x} \in B\} > \beta$, so the root cannot be fathomed by bound when the incumbent is $\beta$. Hence the root must branch on some coordinate $x_i$. By assumption (ii) for this $i$, there exist $\mathbf{u}, \mathbf{v} \in B \cap \{0,1\}^d$ with $\langle \mathbf{w}, \mathbf{u} \rangle = \langle \mathbf{w}, \mathbf{v} \rangle = \beta$ and $u_i \neq v_i$. Since $u_i, v_i \in \{0,1\}$ and $u_i \neq v_i$, exactly one of $u_i, v_i$ equals 0 and the other equals 1. Therefore the child $x_i = 0$ contains one of $\mathbf{u}, \mathbf{v}$ and the child $x_i = 1$ contains the other. Hence both children are feasible and $|\mathcal{T}| \geq 3 = 2^2 - 1$.

**Inductive step.** Fix $m \geq 2$ and assume the claim holds for $m - 1$ blocks. Consider any branch-and-bound tree $\mathcal{T}$ that solves the instance with $m$ blocks. The root branches on some coordinate $x_i^{(t)}$ for some $t \in [m]$ and $i \in [d]$, creating two children with $x_i^{(t)} = 0$ and $x_i^{(t)} = 1$. By assumption (ii) applied to coordinate $i$, there exist $\mathbf{u}, \mathbf{v} \in B \cap \{0,1\}^d$ with $\langle \mathbf{w}, \mathbf{u} \rangle = \langle \mathbf{w}, \mathbf{v} \rangle = \beta$ and $u_i \neq v_i$. Without loss of generality, assume $u_i = 0$ and $v_i = 1$.

For each $b \in \{0,1\}$, let $\mathcal{T}^{(b)}$ be the subtree rooted at the child $x_i^{(t)} = b$. We analyze $\mathcal{T}^{(0)}$. The argument for $\mathcal{T}^{(1)}$ is identical after replacing $\mathbf{u}$ by $\mathbf{v}$. Consider the subproblem obtained by additionally fixing block $t$ to $\mathbf{u}$. This restriction is feasible in the child $x_i^{(t)} = 0$ since $u_i = 0$. From $\mathcal{T}^{(0)}$, whenever a node branches on a variable in block $t$, keep only the child consistent with $\mathbf{x}^{(t)} = \mathbf{u}$ and delete the other child subtree. Contract the resulting degree 1 branching nodes on block $t$. The resulting tree is a valid branch-and-bound tree for the restricted instance, it has at most $|\mathcal{T}^{(0)}|$ nodes, and every leaf is still either infeasible or has LP upper bound at most $m\beta$. Let $\widehat{\mathcal{T}}^{(0)}$ denote this contracted tree.

Relabel blocks so that $t = 1$. Under the restriction $\mathbf{x}^{(1)} = \mathbf{u}$, the feasible region becomes $\{\mathbf{u}\} \times B^{m-1}$. Write $\mathbf{x} = (\mathbf{u}, \mathbf{y})$ with $\mathbf{y} = (\mathbf{y}^{(1)}, \dots, \mathbf{y}^{(m-1)})$. The objective on this restricted instance is

$$\sum_{t=1}^m \langle \mathbf{w}, \mathbf{x}^{(t)} \rangle = \langle \mathbf{w}, \mathbf{u} \rangle + \sum_{s=1}^{m-1} \langle \mathbf{w}, \mathbf{y}^{(s)} \rangle = \beta + \sum_{s=1}^{m-1} \langle \mathbf{w}, \mathbf{y}^{(s)} \rangle.$$

Since $\widehat{\mathcal{T}}^{(0)}$ has no branchings on variables in the first block, we may interpret $\widehat{\mathcal{T}}^{(0)}$ as a branch-and-bound tree for the MILP with $m - 1$ blocks

$$\max\left\{\sum_{s=1}^{m-1} \langle \mathbf{w}, \mathbf{y}^{(s)} \rangle : \mathbf{y} \in B^{m-1}, \ \mathbf{y} \in \{0,1\}^{d(m-1)}\right\}$$

with at most $|\widehat{\mathcal{T}}^{(0)}|$ nodes. Since $\mathbf{x}^{(1)}$ is fixed to the integer vector $\mathbf{u}$, its contribution to the objective is the constant $\beta$ in every node relaxation of $\widehat{\mathcal{T}}^{(0)}$. Thus, every leaf of $\widehat{\mathcal{T}}^{(0)}$ has LP upper bound at most $(m - 1)\beta$ when interpreted as a branch-and-bound tree for the MILP with $m - 1$ blocks. Moreover, every integer feasible solution to the instance with $m - 1$ blocks has value at most $(m - 1)\beta$, and $(\mathbf{x}^\star, \dots, \mathbf{x}^\star) \in B^{m-1} \cap \{0,1\}^{d(m-1)}$ attains value $(m - 1)\beta$. Thus $\widehat{\mathcal{T}}^{(0)}$ solves the instance with $m - 1$ blocks. Therefore $|\widehat{\mathcal{T}}^{(0)}| \geq 2^m - 1$ by the induction hypothesis. Since $|\mathcal{T}^{(0)}| \geq |\widehat{\mathcal{T}}^{(0)}|$, we have $|\mathcal{T}^{(0)}| \geq 2^m - 1$. The same argument shows $|\mathcal{T}^{(1)}| \geq 2^m - 1$.

Since the root contributes one node and the two subtrees are disjoint,

$$|\mathcal{T}| \geq 1 + |\mathcal{T}^{(0)}| + |\mathcal{T}^{(1)}| \geq 1 + 2(2^m - 1) = 2^{m+1} - 1.$$

$\square$

*Proof of Theorem 4.3.* Fix $\varepsilon > 0$ and $n \geq 4$. Let

$$m = \left\lfloor \frac{n}{3} \right\rfloor, \qquad r = n - 3m \in \{0, 1, 2\}, \qquad \varepsilon' = \min\left\{\frac{1}{4}, \frac{\varepsilon}{m}\right\}.$$

We construct the instance $I$ as follows. The variables are $\mathbf{x} \in \mathbb{R}^{3m}$ and $\mathbf{y} \in \mathbb{R}^r$. We group $\mathbf{x}$ into blocks $\mathbf{x}^{(t)} = (x_1^{(t)}, x_2^{(t)}, x_3^{(t)})$ for $t \in [m]$. The objective is

$$\max \sum_{t=1}^{m} (x_1^{(t)} + x_2^{(t)} + x_3^{(t)}).$$

The constraints are, for each $t \in [m]$,

$$x_1^{(t)} + x_2^{(t)} \leq 1, \qquad x_1^{(t)} + x_3^{(t)} \leq 1, \qquad x_2^{(t)} + x_3^{(t)} \leq 1, \qquad 0 \leq x_i^{(t)} \leq 1 \text{ for } i \in [3],$$

together with $\mathbf{y} = \mathbf{0}$ and integrality $\mathbf{x} \in \{0, 1\}^{3m}$. The integer optimum equals $m$, since each block encodes a stable set in a triangle and contributes at most 1 to the objective.

We define the cut sets as follows:

$$\mathcal{C} = \left\{ x_1^{(t)} + x_2^{(t)} + x_3^{(t)} \leq 1 : t \in [m] \right\} \qquad \text{and} \qquad \widetilde{\mathcal{C}} = \left\{ x_1^{(t)} + x_2^{(t)} + x_3^{(t)} \leq 1 + \varepsilon' : t \in [m] \right\}.$$

Both are valid cutting planes: for any integer feasible point, the three pairwise constraints force $x_1^{(t)} + x_2^{(t)} + x_3^{(t)} \leq 1$ in every block $t$. Moreover, the root LP optimum sets $x_1^{(t)} = x_2^{(t)} = x_3^{(t)} = 1/2$ in every block, so both cut sets are violated at the root.

For item 1. in the statement of Theorem 4.3, each cut in $\mathcal{C}$ and its counterpart in $\widetilde{\mathcal{C}}$ have identical coefficients, and their right-hand sides differ by $\varepsilon'$.

For item 2. in the statement of Theorem 4.3, consider the polytope for one block

$$P = \left\{ \mathbf{x} \in [0, 1]^3 : x_1 + x_2 \leq 1, \ x_1 + x_3 \leq 1, \ x_2 + x_3 \leq 1 \right\}.$$

Summing the three inequalities gives $2(x_1 + x_2 + x_3) \leq 3$, so $\max_{\mathbf{x} \in P}(x_1 + x_2 + x_3) = 3/2$, achieved at $(1/2, 1/2, 1/2)$.

With the cut set $\mathcal{C}$, each block also satisfies $x_1 + x_2 + x_3 \leq 1$, and the block LP value becomes 1. With the cut set $\widetilde{\mathcal{C}}$, we obtain $x_1 + x_2 + x_3 \leq 1 + \varepsilon'$, and the block LP value becomes $1 + \varepsilon'$. Indeed, the constraint gives the upper bound, and the point $(1 - \varepsilon', \varepsilon', \varepsilon')$ is feasible since $2\varepsilon' \leq 1$ and has objective $1 + \varepsilon'$.

Thus,

$$z_{\mathrm{LP}}(I; \mathcal{C}) = m \qquad \text{and} \qquad z_{\mathrm{LP}}(I; \widetilde{\mathcal{C}}) = m(1 + \varepsilon').$$

Therefore $|z_{\mathrm{LP}}(I; \mathcal{C}) - z_{\mathrm{LP}}(I; \widetilde{\mathcal{C}})| = m\varepsilon' \leq \varepsilon$.

For item 3. in the statement of Theorem 4.3, the root LP value is $z_{\mathrm{LP}}(I) = \frac{3m}{2}$ and the mixed-integer optimum is $m$, so the root LP gap equals $\frac{m}{2}$. With $\mathcal{C}$, the LP value becomes $m$, so the closed fraction is 1. With $\widetilde{\mathcal{C}}$, the LP value becomes $m(1 + \varepsilon')$, so the remaining gap is $m\varepsilon'$ and the closed fraction is $1 - \frac{m\varepsilon'}{m/2} = 1 - 2\varepsilon'$.

For item 4. in the statement of Theorem 4.3, consider first $\mathcal{C}$. Under $\mathcal{C}$, the root LP value equals $m$. Since there is an integer feasible solution of value $m$, the root already certifies optimality and $|\mathcal{T}(I; \mathcal{C})| = 1$.

For $\widetilde{\mathcal{C}}$, define the polytope for one block

$$B_{\varepsilon'} = \left\{ \mathbf{x} \in [0, 1]^3 : x_1 + x_2 \leq 1, \ x_1 + x_3 \leq 1, \ x_2 + x_3 \leq 1, \ x_1 + x_2 + x_3 \leq 1 + \varepsilon' \right\}.$$

Let $\mathbf{w} = \mathbf{1} \in \mathbb{R}^3$. Let $\beta = 1$. Every point in $B_{\varepsilon'} \cap \{0, 1\}^3$ is either $\mathbf{0}$ or a standard basis vector, so $\max\left\{ \langle \mathbf{w}, \mathbf{x} \rangle : \mathbf{x} \in B_{\varepsilon'} \cap \{0, 1\}^3 \right\} = \beta$. Moreover, $\max\left\{ \langle \mathbf{w}, \mathbf{x} \rangle : \mathbf{x} \in B_{\varepsilon'} \right\} = 1 + \varepsilon'$ by the computation for one block

above. Finally, fix any $i \in [3]$ and take $\mathbf{u} = \mathbf{e}_i$ and $\mathbf{v} = \mathbf{e}_j$ for any $j \neq i$. Then $\mathbf{u}, \mathbf{v} \in B_{\varepsilon'} \cap \{0,1\}^3$, $\langle \mathbf{w}, \mathbf{u} \rangle = \langle \mathbf{w}, \mathbf{v} \rangle = \beta$, and $u_i \neq v_i$.

Applying Lemma 4.2 to $B_{\varepsilon'}$ and $\mathbf{w}$ shows that any branch-and-bound tree that solves the 0-1 MILP

$$\max \left\{ \sum_{t=1}^{m} (x_1^{(t)} + x_2^{(t)} + x_3^{(t)}) : \mathbf{x} \in B_{\varepsilon'}^m, \ \mathbf{x} \in \{0,1\}^{3m} \right\}$$

has at least $2^{m+1} - 1$ nodes. Since $\mathbf{y}$ is fixed to $\mathbf{0}$, the same lower bound holds for $I$ after adding $\widetilde{\mathcal{C}}$. Therefore $|\mathcal{T}(I; \widetilde{\mathcal{C}})| \geq 2^{m+1} - 1$.

For the matching upper bound, consider the complete binary tree of depth $m$ obtained by branching on the variables $x_1^{(1)}, x_1^{(2)}, \ldots, x_1^{(m)}$ in this order along every root to leaf path. At any leaf, for each block $t$ we have either $x_1^{(t)} = 1$, which forces $x_2^{(t)} = x_3^{(t)} = 0$, or $x_1^{(t)} = 0$, which forces $x_2^{(t)} + x_3^{(t)} \leq 1$. In both cases, the block contributes at most 1 to the objective, so $\sum_{t=1}^{m} (x_1^{(t)} + x_2^{(t)} + x_3^{(t)}) \leq m$ holds on every leaf region. At any node where $x_1^{(t)}$ is not yet fixed, the block LP optimum attains value $1 + \varepsilon'$ and in particular has $x_1^{(t)}$ fractional, so each branching variable is fractional at the node LP optimum. This gives a branch-and-bound tree with exactly $2^{m+1} - 1$ nodes, hence $|\mathcal{T}(I; \widetilde{\mathcal{C}})| \leq 2^{m+1} - 1$.

Combining the bounds yields $|\mathcal{T}(I; \widetilde{\mathcal{C}})| = 2^{m+1} - 1 = 2^{\lfloor n/3 \rfloor + 1} - 1$. $\qquad\square$

## B. Proofs for Branching Variable Selection

### B.1. Setup and the Instance Family

We consider 0–1 MILPs in packing form

$$\max \left\{ \mathbf{c}^\top \mathbf{x} : A\mathbf{x} \leq \mathbf{b}, \ \mathbf{x} \in \{0,1\}^{n_{\text{var}}} \right\}. \tag{3}$$

We use best bound node selection and branching on LP fractional variables, as described in Section 3. Strong branching selects the variable maximizing the product score (2), with ties broken by smallest index.

**Definition B.1** (The instance family). Fix an integer $n \in \mathbb{N}$, and set $M = 7n + 8$. The instance $I_n$ is the binary packing problem (3) defined as follows.

For each $i \in [n]$, introduce variables $b_i, p_i, y_{i,1}, y_{i,2}, y_{i,3} \in \{0,1\}$. We fix the variable ordering

$$\mathbf{x} = (b_1, \ldots, b_n, \ p_1, \ldots, p_n, \ y_{1,1}, \ldots, y_{n,1}, \ y_{1,2}, \ldots, y_{n,2}, \ y_{1,3}, \ldots, y_{n,3}).$$

For each $i \in [n]$, impose the four inequalities

$$2b_i + p_i \leq 2, \tag{4}$$
$$b_i + y_{i,1} + y_{i,2} \leq 2, \tag{5}$$
$$b_i + y_{i,1} + y_{i,3} \leq 2, \tag{6}$$
$$b_i + y_{i,2} + y_{i,3} \leq 2. \tag{7}$$

The objective is

$$\max \sum_{i=1}^{n} \left( 24b_i + Mp_i + 4y_{i,1} + 8y_{i,2} + 8y_{i,3} \right). \tag{8}$$

We will use two auxiliary LP calculations for a single block of the instance family. We analyze a single block, dropping the block index $i$ and writing $(b, p, y_1, y_2, y_3) \in [0,1]^5$ with objective $24b + Mp + 4y_1 + 8y_2 + 8y_3$ and constraints (4)–(7).

**Lemma B.2.** *For $t \in [0, 2]$, consider the polyhedron*

$$Y(t) = \left\{ \mathbf{y} \in [0,1]^3 : y_1 + y_2 \leq t, \ y_1 + y_3 \leq t, \ y_2 + y_3 \leq t \right\}.$$

*Then*

$$\max_{\mathbf{y} \in Y(t)} 4y_1 + 8y_2 + 8y_3 = 10t,$$

*and the unique maximizer is $y_1 = y_2 = y_3 = t/2$.*

*Proof.* The point $(t/2, t/2, t/2) \in [0,1]^3$ is feasible for $t \le 2$ and has value $10t$, so the maximum is at least $10t$.

Let $\mathbf{y} \in Y(t)$ be arbitrary. Then

$$4y_1 + 8y_2 + 8y_3 = 2(y_1 + y_2) + 2(y_1 + y_3) + 6(y_2 + y_3)$$
$$\le 2t + 2t + 6t$$
$$= 10t.$$

Hence the maximum is at most $10t$, and therefore it equals $10t$.

If $\mathbf{y}$ is optimal, then $4y_1 + 8y_2 + 8y_3 = 10t$, so the inequalities above hold with equality. Since $y_1 + y_2 \le t$, $y_1 + y_3 \le t$, and $y_2 + y_3 \le t$, this implies $y_1 + y_2 = t$, $y_1 + y_3 = t$, and $y_2 + y_3 = t$. This linear system has the unique solution $y_1 = y_2 = y_3 = t/2$. $\square$

**Lemma B.3.** *Let $n \in \mathbb{N}$, let $M = 7n + 8$, and consider a single block of $I_n$. In each item below, the stated LP value is computed under the specified fixings, with all other variables in the block left free.*

1. *If no variables in the block are fixed, then the LP relaxation has the unique optimum*

$$b = \frac{1}{2}, \qquad p = 1, \qquad y_1 = y_2 = y_3 = \frac{3}{4},$$

   *with LP value $M + 27$.*

2. *If $b$ is fixed to $0$, then the block LP value is $M + 20$. If $b$ is fixed to $1$, then the block LP value is $34$.*

3. *If $y_1$ is fixed to $0$ or to $1$, then the block LP value is $M + 24$ in both cases.*

4. *If $y_2$ is fixed to $0$, then the block LP value is $M + 22$. If $y_2$ is fixed to $1$, then the block LP value is $M + 26$. The same conclusions hold when $y_3$ is fixed instead of $y_2$.*

*Proof.* (1) Fix $b \in [0,1]$. Maximizing the objective subject to $2b + p \le 2$ and $p \in [0,1]$ sets $p = \min\{1, 2 - 2b\}$. Let $t = 2 - b \in [1, 2]$. Constraints (5)–(7) are equivalent to $\mathbf{y} \in Y(t)$, so Lemma B.2 yields

$$\max_{\mathbf{y} \in Y(t)} (4y_1 + 8y_2 + 8y_3) = 10t,$$

attained uniquely at $y_1 = y_2 = y_3 = t/2$. Therefore, for fixed $b$ the block LP value equals

$$g(b) = 24b + M \min\{1, 2 - 2b\} + 10(2 - b).$$

If $b \le 1/2$ then $g(b) = M + 20 + 14b$, which is maximized on $[0, 1/2]$ at $b = 1/2$. If $b \ge 1/2$ then $g(b) = 2M + 20 + (14 - 2M)b$, which is maximized on $[1/2, 1]$ at $b = 1/2$ since $M \ge 15$. Thus $b = 1/2$ is the unique maximizer, and the corresponding optimum is $p = 1$ and $y_1 = y_2 = y_3 = 3/4$, yielding LP value $M + 27$.

(2) If $b = 0$, then (4) permits $p = 1$, and (5)–(7) permit $y_1 = y_2 = y_3 = 1$. This yields value $M + 20$ and is optimal since all objective coefficients are nonnegative. If $b = 1$, then (4) forces $p = 0$ and (5)–(7) reduce to $Y(1)$. By Lemma B.2, the maximum $y$-value is $10$, so the value is $24 + 10 = 34$.

(3) We first treat $y_1 = 0$. Constraints (5) and (6) become $b + y_2 \le 2$ and $b + y_3 \le 2$, and (7) becomes $b + y_2 + y_3 \le 2$. Fix $b \in [0, 1]$. Maximizing over $p$ again gives $p = \min\{1, 2 - 2b\}$. The remaining constraints imply $y_2, y_3 \in [0, 1]$ and $y_2 + y_3 \le 2 - b$, so the maximum of $8y_2 + 8y_3$ equals $8(2 - b)$, attained whenever $y_2 + y_3 = 2 - b$. Hence the block LP value for fixed $b$ equals

$$h_0(b) = 24b + M \min\{1, 2 - 2b\} + 8(2 - b).$$

If $b \le 1/2$ then $h_0(b) = M + 16 + 16b$. If $b \ge 1/2$ then $h_0(b) = 2M + 16 + (16 - 2M)b$. In both cases the maximum is attained uniquely at $b = 1/2$, yielding $p = 1$ and block LP value $M + 24$. One optimal choice sets $y_2 = y_3 = 3/4$.

Now treat $y_1 = 1$. Constraints (5) and (6) become $b + y_2 \leq 1$ and $b + y_3 \leq 1$, and (7) becomes $b + y_2 + y_3 \leq 2$. Fix $b \in [0,1]$. Maximizing over $p$ gives $p = \min\{1, 2 - 2b\}$. The first two inequalities imply $y_2, y_3 \leq 1 - b$, so the maximum of $8y_2 + 8y_3$ equals $16(1 - b)$, attained uniquely at $y_2 = y_3 = 1 - b$. Hence the block LP value for fixed $b$ equals

$$h_1(b) = 24b + M \min\{1, 2 - 2b\} + 4 + 16(1 - b).$$

If $b \leq 1/2$ then $h_1(b) = M + 20 + 8b$. If $b \geq 1/2$ then $h_1(b) = 2M + 20 + (8 - 2M)b$. In both cases the maximum is attained uniquely at $b = 1/2$, yielding $p = 1$, $y_2 = y_3 = 1/2$, and block LP value $M + 24$.

(4) We analyze branching on $y_2$; the case with $y_3$ is the same by symmetry. Fix $y_2 = 0$ and assume no other variables in the block are fixed. Then (5) becomes $b + y_1 \leq 2$ and (7) becomes $b + y_3 \leq 2$, while (6) becomes $b + y_1 + y_3 \leq 2$. Fix $b \in [0,1]$. Maximizing over $p$ gives $p = \min\{1, 2 - 2b\}$. The remaining constraints imply $y_1, y_3 \in [0,1]$ and $y_1 + y_3 \leq 2 - b$, so the maximum of $4y_1 + 8y_3$ equals $4(1 - b) + 8$, attained at $y_3 = 1$ and $y_1 = 1 - b$. Therefore the block LP value for fixed $b$ equals

$$24b + M \min\{1, 2 - 2b\} + 4(1 - b) + 8.$$

If $b \leq 1/2$ then this equals $M + 12 + 20b$, which is maximized on $[0, 1/2]$ at $b = 1/2$. If $b \geq 1/2$ then this equals $2M + 12 + (20 - 2M)b$, which is maximized on $[1/2, 1]$ at $b = 1/2$ since $M \geq 15$. Thus the block LP value is $M + 22$.

Next fix $y_2 = 1$. Then (5) becomes $b + y_1 \leq 1$ and (7) becomes $b + y_3 \leq 1$, while (6) becomes $b + y_1 + y_3 \leq 2$. Fix $b \in [0,1]$ and maximize over $p$ to get $p = \min\{1, 2 - 2b\}$. The first two inequalities imply $y_1, y_3 \leq 1 - b$, so the maximum of $4y_1 + 8y_3$ equals $12(1 - b)$, attained uniquely at $y_1 = y_3 = 1 - b$. Therefore the block LP value for fixed $b$ equals

$$24b + M \min\{1, 2 - 2b\} + 8 + 12(1 - b).$$

If $b \leq 1/2$ then this equals $M + 20 + 12b$, which is maximized on $[0, 1/2]$ at $b = 1/2$. If $b \geq 1/2$ then this equals $2M + 20 + (12 - 2M)b$, which is maximized on $[1/2, 1]$ at $b = 1/2$ since $M \geq 15$. Thus the block LP value is $M + 26$. $\square$

**Lemma B.4** (Strong branching tree size on $I_n$). *Fix $n \in \mathbb{N}$ and let $I_n$ be the instance from Definition B.1 with parameter $M = 7n + 8$. Under the standing assumptions, the mixed-integer optimum satisfies*

$$\mathrm{OPT}(I_n) = n(M + 20),$$

*and the strong branching policy $\pi_{\mathrm{SB}}$ satisfies*

$$|\mathcal{T}_{\pi_{\mathrm{SB}}}(I_n)| = 2n + 1.$$

*Proof.* We first note that $\mathrm{OPT}(I_n) = n(M + 20)$. Indeed, setting $b_i = 0$ and $p_i = y_{i,1} = y_{i,2} = y_{i,3} = 1$ in every block yields an integer feasible solution of value $n(M + 20)$. If instead $b_i = 1$ in some block, then (4) forces $p_i = 0$ and (5)–(7) imply that at most one of $y_{i,1}, y_{i,2}, y_{i,3}$ can be 1, so the block value is at most $24 + 8 = 32$. Since $M = 7n + 8 \geq 15$, we have $M + 20 > 32$, so an optimal solution sets $b_i = 0$ for all $i$.

We now analyze the strong branching run with best bound node selection. At any node that contains a free block, Lemma B.3 implies that the LP optimum in that block has $b_i = 1/2$ and $y_{i,1} = y_{i,2} = y_{i,3} = 3/4$. Branching on $b_i$ yields LP improvements 7 and $M - 7$, while branching on $y_{i,1}$ yields improvements 3 and 3, and branching on $y_{i,2}$ or $y_{i,3}$ yields improvements 5 and 1. Thus, for any $\eta_{\mathrm{SB}} > 0$, the strong branching product score (2) selects a $b$ variable from a free block. When $\eta_{\mathrm{SB}} \geq M - 7$ this selection can be a tie, since then every candidate in a free block has score $\eta_{\mathrm{SB}}^2$, and the tie-breaking rule selects the smallest index. In particular, at the root the policy branches on $b_1$.

Let $N_t$ denote the node with $b_1 = \cdots = b_t = 0$ and all other variables unfixed. By block separability and Lemma B.3, the LP value of $N_t$ equals $n(M + 27) - 7t$. Any other open node created before depth $n$ has at least one index $i$ with $b_i = 1$ and therefore has LP value at most $n(M + 27) - (M - 7)$. Since $M = 7n + 8$, for all $t \leq n - 1$ we have $n(M + 27) - 7t > n(M + 27) - (M - 7)$, so best bound always selects $N_t$ next. At depth $n$, node $N_n$ is integral and has value $n(M + 20) = \mathrm{OPT}(I_n)$. All remaining open nodes have LP value at most $n(M + 27) - (M - 7) < \mathrm{OPT}(I_n)$ and are pruned by bound.

Counting created nodes gives one root and two children for each of the $n$ branchings, so $|\mathcal{T}_{\pi_{\mathrm{SB}}}(I_n)| = 2n + 1$. $\square$

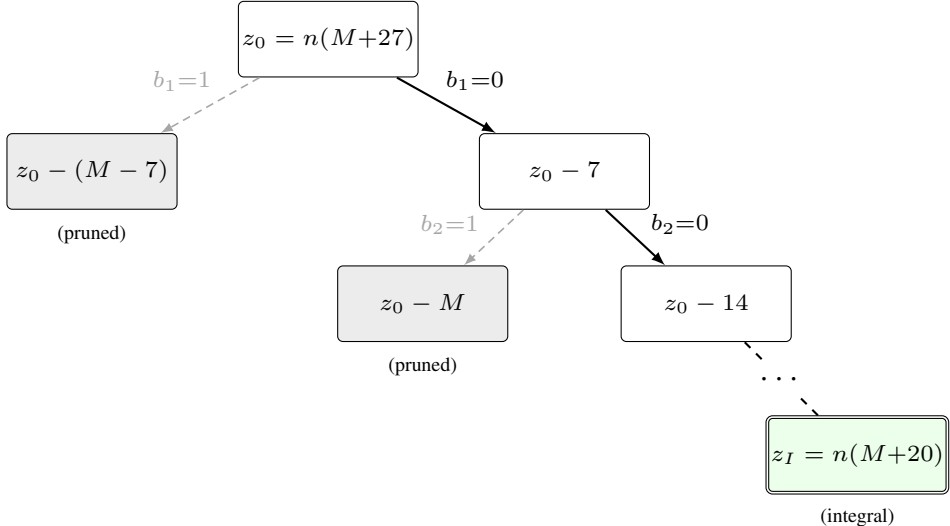

*Figure 6.* The tree under $\pi_{\mathrm{SB}}$ on $I_n$. Node labels show LP bounds. Here $z_0 = n(M{+}27)$ is the root LP value and $z_I = n(M{+}20)$ is the incumbent found at depth $n$. Best bound follows the chain $b_1 = \cdots = b_n = 0$, and siblings with $b_i = 1$ are pruned by bound.

## B.2. Proof of Theorem 5.1

*Proof of Theorem 5.1.* Fix $n \geq 3$ and $\varepsilon > 0$. Let $M = 7n + 8$, and let $\eta_{\mathrm{SB}} > 0$ be the constant in the strong branching score (2). Let $I_n$ be the instance from Definition B.1 with parameter $n$ and objective (8). Define the scaling factor

$$\alpha := \frac{\eta_{\mathrm{SB}}}{2(M + 27)}.$$

Let $\widetilde{I}_n$ be the instance obtained from $I_n$ by multiplying every objective coefficient by $\alpha$. This does not change the feasible region, the variable ordering, or the set of LP optimal solutions at any node. Moreover, every node LP value and every LP bound improvement scale by $\alpha$.

We use the block LP calculations from Lemma B.3 for $I_n$. After scaling, a free block has LP value $\alpha(M + 27)$ and an optimal LP solution with $b = \frac{1}{2}$, $p = 1$, and $y_1 = y_2 = y_3 = \frac{3}{4}$. If $b$ is fixed to 0 or 1, the block LP values are $\alpha(M + 20)$ and $\alpha \cdot 34$, respectively. If $y_1$ is fixed to 0 or 1, the block LP value is $\alpha(M + 24)$ in both cases.

We first show that strong branching scores are constant on $\widetilde{I}_n$. Consider any node $N$ in the branch-and-bound tree generated by either policy constructed below, and fix any candidate branching variable $j$ at $N$. Let $i$ be the block containing $j$, and for each block $t$ let $v_t(N)$ denote the optimal value of the LP relaxation of block $t$ at $N$. By separability across blocks,

$$z(N) = \sum_{t=1}^{n} v_t(N).$$

For $b \in \{0, 1\}$, the child $N^{(b)}$ differs from $N$ only through additional fixings in block $i$, hence

$$z(N^{(b)}) = v_i(N^{(b)}) + \sum_{t \neq i} v_t(N),$$

and therefore $\Delta^{(b)}(N, j) = v_i(N) - v_i(N^{(b)})$.

We claim that both children $N^{(0)}$ and $N^{(1)}$ are LP feasible. For the branch $j = 0$, take any LP feasible solution at $N$ and set the variable indexed by $j$ to 0. This preserves feasibility since all constraints have nonnegative coefficients. For the branch $j = 1$, we give an explicit feasible assignment for block $i$ that is consistent with the fixings at $N$. If the variable indexed by $j$ is $b_i$, set $b_i = 1$ and $p_i = 0$, and set every $y$ variable in that block that is not fixed at $N$ to 0. Otherwise, if the variable indexed by $j$ is $p_i$, note that $b_i$ cannot be fixed to 1 at $N$, since $2b_i + p_i \leq 2$ would force $p_i = 0$. Set $p_i = 1$ and $b_i = 0$, and set every $y$ variable in that block that is not fixed at $N$ to 0. Otherwise, the variable indexed by $j$ is a $y$ variable in block $i$.

Set it to 1 and set the other two $y$ variables that are not fixed at $N$ to 0. If $b_i$ is fixed to 1 at $N$, set $p_i = 0$, and otherwise set $b_i = 0$ and $p_i = 1$. In every case the block constraints (4)–(7) are satisfied, so both children are feasible.

Since all objective coefficients are nonnegative, we have $v_i(N^{(b)}) \geq 0$ for $b \in \{0, 1\}$, and hence $\Delta^{(b)}(N, j) \leq v_i(N)$. Moreover, each block has LP value at most $\alpha(M + 27)$ at every node, because a free block attains LP value $\alpha(M + 27)$ and additional fixings can only decrease the LP value. Hence,

$$\Delta^{(0)}(N, j) \leq \alpha(M + 27) = \frac{\eta_{\text{SB}}}{2} \qquad \text{and} \qquad \Delta^{(1)}(N, j) \leq \alpha(M + 27) = \frac{\eta_{\text{SB}}}{2}.$$

In particular, $\Delta^{(0)}(N, j) < \eta_{\text{SB}}$ and $\Delta^{(1)}(N, j) < \eta_{\text{SB}}$. Therefore (2) implies

$$\text{Score}_{\text{SB}}(N, j) = \eta_{\text{SB}}^2 \qquad \text{for every such node } N \text{ and every candidate variable } j.$$

We now bound the tree size under strong branching. Let $\pi_{\text{SB}}$ be the strong branching policy, with ties broken by the smallest index under the variable ordering in Definition B.1. Since $\text{Score}_{\text{SB}}(N, j)$ is constant over candidates, $\pi_{\text{SB}}$ always branches on the smallest index variable that is fractional in an optimal LP solution at the current node. At any node that contains a free block, Lemma B.3(1) implies that $b_i = 1/2$ in the unique block LP optimum, so $\pi_{\text{SB}}$ branches on the smallest index free $b_i$. Moreover, scaling the objective by $\alpha$ multiplies every node LP value by the same factor and therefore preserves the node ordering under best bound. Thus, the best bound comparisons in the proof of Lemma B.4 carry over directly after scaling, and we obtain

$$|\mathcal{T}_{\pi_{\text{SB}}}(\widetilde{I}_n)| = 2n + 1.$$

As in the proof of Lemma B.4, we have $\text{OPT}(I_n) = n(M + 20)$ and therefore $\text{OPT}(\widetilde{I}_n) = \alpha n(M + 20)$.

We now define a policy whose scores are $\varepsilon$ close to $\text{Score}_{\text{SB}}$ and that generates an exponential tree. Define a scoring function $\widehat{\text{Score}}$ by

$$\widehat{\text{Score}}(N, j) = \begin{cases} \text{Score}_{\text{SB}}(N, j) + \varepsilon/2, & \text{if } j \text{ is the index of } y_{i,1} \text{ for some } i \text{ and } y_{i,1} \text{ is fractional at } N, \\ \text{Score}_{\text{SB}}(N, j), & \text{otherwise.} \end{cases}$$

Then for every node $N$ and every index $j$,

$$\left| \widehat{\text{Score}}(N, j) - \text{Score}_{\text{SB}}(N, j) \right| \leq \varepsilon/2 \leq \varepsilon,$$

which proves item 1 in the theorem statement.

Let $\hat{\pi}$ be the argmax policy induced by $\widehat{\text{Score}}$, with ties broken by the smallest index. At the root, every block is free, hence $y_{1,1}$ is fractional in the root LP optimum. Since $\text{Score}_{\text{SB}}$ is constant over candidates, every fractional $y_{i,1}$ has score $\eta_{\text{SB}}^2 + \varepsilon/2$, while every other candidate has score $\eta_{\text{SB}}^2$. Therefore $\hat{\pi}$ branches on $y_{1,1}$.

More generally, consider a node $N$ at depth $d \leq n$ in the tree generated by $\hat{\pi}$. Such a node is obtained by fixing $y_{1,1}, \ldots, y_{d,1}$, each to 0 or 1, and leaving all $b$ variables unfixed. By separability across blocks and the scaled block values above, the first $d$ blocks contribute $\alpha(M + 24)$ each, and the remaining $n - d$ blocks are free and contribute $\alpha(M + 27)$ each. Hence

$$z(N) = d\alpha(M + 24) + (n - d)\alpha(M + 27) = \alpha\big(n(M + 27) - 3d\big).$$

In particular, for $d \leq n$,

$$z(N) \geq \alpha n(M + 24) > \alpha n(M + 20) = \text{OPT}(\widetilde{I}_n),$$

and since every incumbent value is at most $\text{OPT}(\widetilde{I}_n)$, no such node can ever be pruned by bound.

If $d < n$, then block $d + 1$ is free. By Lemma B.3(1), the unique optimal LP solution in that block has $y_{d+1,1} = \frac{3}{4}$, so $y_{d+1,1}$ is a candidate branching variable. By construction of $\widehat{\text{Score}}$, every fractional $y_{i,1}$ has strictly larger score than every other candidate, so $\hat{\pi}$ branches on $y_{d+1,1}$ at every node of depth $d$. Fixing $y_{d+1,1}$ to 0 or 1 yields two LP feasible children, since the block LP value equals $\alpha(M + 24)$ in both cases. Also, since $d < n$ at least one block is free, its LP optimum has $b = \frac{1}{2}$ and the node is not integral. Thus every node of depth $d < n$ is branched and has two feasible children.

It follows that the tree generated by $\hat{\pi}$ contains the full binary tree of depth $n$. Hence

$$|\mathcal{T}_{\hat{\pi}}(\widetilde{I}_n)| \geq \sum_{d=0}^{n} 2^d = 2^{n+1} - 1,$$

which proves item 2 and completes the proof. $\qquad\square$

## B.3. Proof of Proposition 5.2

*Proof of Proposition 5.2.* Fix $n \geq 3$. Let $I_n$ be the instance from Definition B.1 with parameter $n$. Fix any $\kappa \in (0, 9]$. Define the scoring function

$$\text{Score}(N, j) = \min \left\{ \text{Score}_{\text{SB}}(N, j), \kappa \right\}.$$

Consider any node $N$ that contains a free block $i$. By Lemma B.3(1), in a free block the unique LP optimum has $b_i = \frac{1}{2}$ and $y_{i,1} = \frac{3}{4}$, so both are fractional and the block LP value equals $M + 27$.

Let $j_y$ and $j_b$ denote the indices of $y_{i,1}$ and $b_i$. Branching on $j_y$ fixes $y_{i,1} \in \{0, 1\}$. By Lemma B.3(3), both children have block LP value $M + 24$. Hence the two LP improvements are 3 and 3, and $\text{Score}_{\text{SB}}(N, j_y) \geq 9$. Branching on $j_b$ fixes $b_i \in \{0, 1\}$. By Lemma B.3(2), the child block LP values are $M + 20$ and 34. Hence the two LP improvements are 7 and $M - 7$, and $\text{Score}_{\text{SB}}(N, j_b) > 9$. Since $\kappa \leq 9$, we have $\text{Score}(N, j_y) = \text{Score}(N, j_b) = \kappa$. Thus, whenever a node contains a free block, the argmax set contains both a $b$ variable and a $y_{i,1}$ variable.

Let $\pi_{\min}$ be the argmax policy induced by Score with ties broken by the smallest index. Let $\pi_y$ be the argmax policy induced by Score with the following tie-breaking rule: among the maximizers, select the smallest index variable of the form $y_{i,1}$ if such a maximizer exists, and otherwise select the smallest index maximizer.

We first analyze $\pi_{\min}$. Since ties are broken by the smallest index and all $b$ variables precede all $y$ variables in the ordering, $\pi_{\min}$ branches on the smallest index fractional $b_i$ whenever a node contains a free block. In particular, $\pi_{\min}$ branches on $b_1$ at the root. The remainder of the argument is the same as in the proof of Lemma B.4: best bound follows the chain $b_1 = \cdots = b_n = 0$, finds an optimal incumbent at depth $n$, and prunes all remaining open nodes by bound. Therefore $|\mathcal{T}_{\pi_{\min}}(I_n)| = 2n + 1$.

Next we analyze $\pi_y$. We claim that at every node at depth $d < n$ in the tree generated by $\pi_y$, the selected branching variable is $y_{d+1,1}$. After $d$ branchings, the variables $y_{1,1}, \ldots, y_{d,1}$ are fixed, and block $d + 1$ is free. Therefore $y_{d+1,1}$ is fractional and belongs to the argmax set, so the tie-breaking rule selects it. Thus $\pi_y$ branches on $y_{1,1}, y_{2,1}, \ldots, y_{n,1}$ along each root to leaf path until depth $n$. As in the proof of Lemma B.4, we have $\text{OPT}(I_n) = n(M + 20)$. For any node at depth $d \leq n$ in the tree generated by $\pi_y$, the first $d$ blocks contribute $M + 24$ each, and the remaining $n - d$ blocks are free and contribute $M + 27$ each. Thus the node LP value equals $n(M + 27) - 3d \geq n(M + 24) > \text{OPT}(I_n)$, so no such node can be pruned by bound. Moreover, for $d < n$ at least one block is free, so the node is not integral and branching on $y_{d+1,1}$ produces two LP feasible children by Lemma B.3(3). Hence the run generates the full binary tree of depth $n$, and

$$|\mathcal{T}_{\pi_y}(I_n)| \geq 2^{n+1} - 1.$$

$\square$

## B.4. Proof of Theorem 5.4

*Proof of Theorem 5.4.* Fix $n \in \mathbb{N}$ and $k \in \{0, 1, \ldots, n\}$, and consider the instance $I_n$ with parameter $M = 7n + 8$. By Lemma B.4, we have $|\mathcal{T}_{\pi_{\text{SB}}}(I_n)| = 2n + 1$ and $\text{OPT}(I_n) = n(M + 20)$. We construct a policy $\hat{\pi}_{n,k}$ and lower bound $|\mathcal{T}_{\hat{\pi}_{n,k}}(I_n)|$.

Recall that a branch-and-bound node $N$ is associated with the subproblem obtained from the original instance by adding branching fixings. Since $I_n$ is a 0–1 instance, each branching fixing sets one binary variable to 0 or 1. Formally, we represent $N$ by the resulting set of binary fixings. We define $\text{depth}(N)$ as the size of this set, so the node with no branching fixings has depth 0. Since we only branch on variables that are fractional in an optimal LP solution, a binary variable is never fixed twice by branching. Thus $\text{depth}(N)$ is well defined, and it equals the number of branching decisions that define $N$. We define $\hat{\pi}_{n,k}$ by a depth threshold. For every node $N$,

$$\hat{\pi}_{n,k}(N) = \begin{cases} y_{\text{depth}(N)+1,1}, & \text{depth}(N) < k, \\ \pi_{\text{SB}}(N), & \text{depth}(N) \geq k. \end{cases}$$

We first verify item (1) in the theorem statement in the sense of Definition 5.3. Along the strong branching run on $I_n$, the internal nodes are exactly

$$N_d: \quad b_1 = \cdots = b_d = 0, \text{ all other variables unfixed}, \quad d = 0, 1, \ldots, n - 1.$$

Moreover, $\text{depth}(N_d) = d$, and by Lemma B.4 we have $\pi_{\text{SB}}(N_d) = b_{d+1}$. For $d < k$, the definition of $\hat{\pi}_{n,k}$ gives $\hat{\pi}_{n,k}(N_d) = y_{d+1,1} \neq b_{d+1}$. For $d \geq k$, we have $\hat{\pi}_{n,k}(N_d) = \pi_{\text{SB}}(N_d)$. Therefore $\hat{\pi}_{n,k}$ differs from $\pi_{\text{SB}}$ on exactly the nodes $N_0, \ldots, N_{k-1}$.

We now lower bound the tree size. Write $\text{OPT} = \text{OPT}(I_n)$. Since the incumbent value is always at most OPT, no node with LP value strictly larger than OPT can be pruned by bound. We claim that every node at depth $d < k$ generated by $\hat{\pi}_{n,k}$ has LP value strictly larger than OPT and is not integral. Indeed, such a node fixes $(y_{1,1}, \ldots, y_{d,1}) \in \{0,1\}^d$ and leaves all $b$ variables unfixed. By block separability and Lemma B.3(1), Lemma B.3(3), each of the first $d$ blocks contributes LP value $M + 24$, and each remaining block contributes LP value $M + 27$. Thus

$$z(N) = d(M + 24) + (n - d)(M + 27) = n(M + 27) - 3d > \text{OPT}.$$

Moreover, block $d + 1$ is free, so its unique blockwise LP optimum has $y_{d+1,1} = 3/4$ by Lemma B.3(1). In particular, $\hat{\pi}_{n,k}(N)$ is a valid branching variable and $N$ is not integral. Therefore every node at depth $d < k$ must be branched. Since branching on $y_{d+1,1}$ yields two LP feasible children by Lemma B.3(3), the first $k$ levels form a full binary tree.

For $\sigma \in \{0,1\}^k$, let $N_\sigma$ denote the node at depth $k$ obtained by fixing $y_{i,1} = \sigma_i$ for all $i \in [k]$ and leaving all other variables unfixed. The $2^k$ nodes $N_\sigma$ have disjoint feasible regions, so the subtrees rooted at these nodes are disjoint. Fix $\sigma \in \{0,1\}^k$. At $N_\sigma$, we have

$$z(N_\sigma) = k(M + 24) + (n - k)(M + 27) = n(M + 27) - 3k = \text{OPT} + (7n - 3k) \geq \text{OPT} + 4n.$$

From depth $k$ onward, $\hat{\pi}_{n,k}$ follows $\pi_{\text{SB}}$. We define a path in the subtree rooted at $N_\sigma$. Set $N^{(0)} = N_\sigma$. For $t \geq 0$, if $N^{(t)}$ is an internal node, let $i$ be the block index of the branching variable selected by $\pi_{\text{SB}}$ at $N^{(t)}$, and define $N^{(t+1)}$ as follows. If the branching variable is $b_i$, then $N^{(t+1)}$ is the child with $b_i = 0$. If the branching variable is $p_i$, then $N^{(t+1)}$ is the child with $p_i = 1$. If the branching variable is a $y$ variable in block $i$, then $N^{(t+1)}$ is either child.

We claim that for every $t \geq 0$ for which $N^{(t)}$ is defined,

$$z(N^{(t)}) \geq z(N_\sigma) - 27t.$$

To see this, note that each branching fixes a variable in a single block, so only that block can change its contribution to the node LP value. The block LP value is at most $M + 27$, since a free block attains LP value $M + 27$ by Lemma B.3(1) and additional fixings can only tighten the feasible region. Let $i$ be the block index selected at $N^{(t)}$. By the definition of the path, node $N^{(t+1)}$ does not impose $b_i = 1$ and does not impose $p_i = 0$. Therefore $N^{(t+1)}$ contains a feasible solution in which $b_i = 0$ and $p_i = 1$, and all other unfixed variables in block $i$ are set to 0. Since all objective coefficients are nonnegative, this solution has block value at least $M$. Therefore the node LP value drops by at most $(M + 27) - M = 27$ in one step, which proves the claim.

Let $T = \lfloor n/7 \rfloor + 1$. For every $t \in \{0, 1, \ldots, T - 1\}$, we have

$$z(N^{(t)}) \geq \text{OPT} + 4n - 27t \geq \text{OPT} + \left(4 - \tfrac{27}{7}\right)n = \text{OPT} + \tfrac{1}{7}n > \text{OPT},$$

so $N^{(t)}$ cannot be fathomed by bound. Also, since $t \leq T - 1 = \lfloor n/7 \rfloor < n$, along the path from $N_\sigma$ to $N^{(t)}$ we branch on variables from at most $t$ distinct blocks, so there exists a block that is not branched on after reaching $N_\sigma$. In an untouched block, every optimal LP solution has $b_i = 1/2$ by Lemma B.3(1) and Lemma B.3(3). This implies that $N^{(t)}$ is not integral. Consequently, each node $N^{(t)}$ for $t \leq T - 1$ is an internal node of the subtree rooted at $N_\sigma$. Since each internal node in a branch-and-bound tree has exactly two children, a tree with $T$ internal nodes has at least $2T + 1$ total nodes. Since $\lfloor n/7 \rfloor \geq n/7 - 1$, we have $2T + 1 = 2\lfloor n/7 \rfloor + 3 \geq 2n/7$. Thus each subtree rooted at $N_\sigma$ contains at least $2n/7$ nodes. Summing over the $2^k$ disjoint subtrees yields $|\mathcal{T}_{\hat{\pi}_{n,k}}(I_n)| \geq 2^k \cdot 2n/7$. This completes the proof. $\qquad\square$

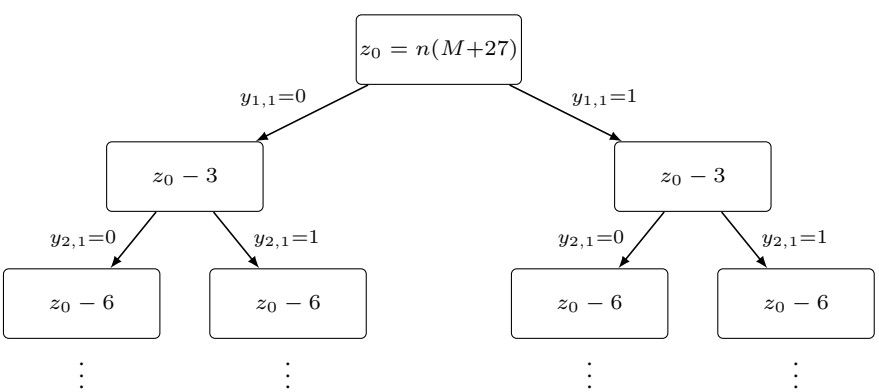

*(a)* Branching on $y_{1,1}, \ldots, y_{k,1}$ (schematic, with only the first two levels shown).

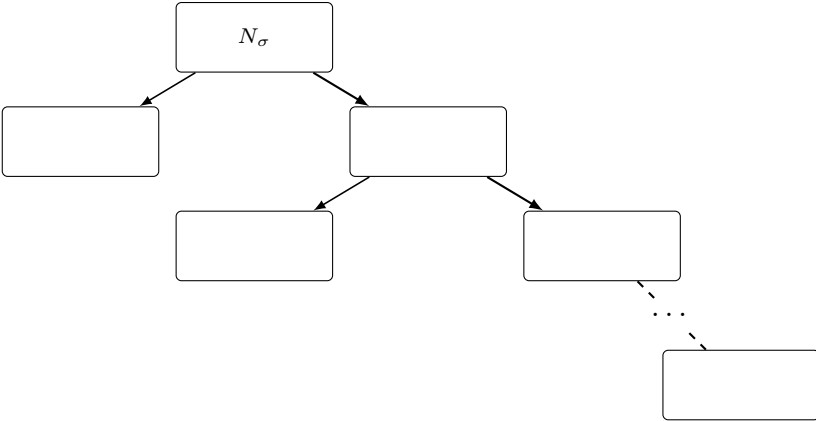

$\Omega(n)$ further branchings

*(b)* Continuing with $\pi_{\mathrm{SB}}$ below $N_\sigma$.

*Figure 7.* Schematic of the tree under $\hat{\pi}_{n,k}$. The first $k$ levels branch on $y_{1,1}, \ldots, y_{k,1}$. Below each of the $2^k$ nodes $N_\sigma$, the policy follows $\pi_{\mathrm{SB}}$ and generates $\Omega(n)$ additional nodes. In particular, $|\mathcal{T}_{\hat{\pi}_{n,k}}(I_n)| \geq 2^k \cdot 2n/7$.

