# OpenReview forum: "Theoretical Challenges in Learning for Branch-and-Cut"
_ICML.cc/2026/Conference — ICML 2026 regular_

### Official Review · Reviewer_EYY5 · 2026-03-12

**Soundness:** 4
**Presentation:** 4
**Significance:** 4
**Originality:** 3
**Overall Recommendation:** 5
**Confidence:** 4

**Summary:**

The authors prove some worst-case suboptimality bounds showing that proxy-driven ML for MILP solvers (common approaches) are vulnerable to substantial performance loss . They prove that (1) a perfect cut-selection expert on LP bound improvement can lead to an exponentially larger tree than a simple heuristic and (2) a small perturbation in the root cut set can make the optimal B&B tree exponentially bigger and also show the vulnerability of mistakes in imitating strong branching: slightly off branching scores can lead to exponentially bigger trees.

**Compliance With Llm Reviewing Policy:**

Affirmed.

**Final Justification:**

I will maintain my already high rating of the paper.

**Key Questions For Authors:**

I do not have critical questions. I do have two questions purely out of interest. The authors' response will not change my evaluation of the paper but will perhaps motivate them to add some commentary.
1. Do you think these results might similarly bite end-to-end approaches?  For example, by learning on these specific constructions, do you think an end-to-end method could potentially learn robust branching/cut strategies which are much less vulnerable to small perturbations? Related: do you think it is possible that for some distributions, a slightly suboptimal policy is always going to have these exponential blowups?
1. These are interesting constructions used to prove your theorems though I have a difficult time intuiting how applicable they are? What is your intuition for how contrived they are? How might we determine if they bite in practice? Do you think these results should be enough to convince ML people that end-to-end approaches have much more upside?

**Limitations:**

yes

**Strengths And Weaknesses:**

The authors take a very impactful problem (customizing MILP solvers  with ML) and show how common successful approaches could break down and potentially leave a lot of performance on the table. They make a good case for end-to-end approaches and this could be very influential for guiding the community going forward. The authors are not the first to explore this area of theory (e.g., Dey, '25), but the authors substantially expand that theory in an interesting way.

The work is presented very clearly. It was comprehensible given it was quite technical and the narrative connecting the various theorems was well designed. This is straightforwardly a very strong paper and I have no major criticisms.

---

> ### Author Rebuttal · Authors · 2026-03-30
>
> Thank you for the generous assessment and the interesting questions.
>
> **Q1.**
>
> This is an excellent question, and we do not yet have a sharp answer. One observation from our branching constructions is that, in the families behind Theorems 5.1 and 5.4, strong branching already gives the optimal tree, yet arbitrarily small perturbations of that rule can still cause exponential blowups. This suggests that even for a globally optimal policy, instability can remain a significant issue.
>
> For this reason, moving from local imitation to an end-to-end method may not by itself remove the phenomenon we highlight here since the end-to-end method may produce optimal sized trees and yet be brittle with respect to some other type of perturbations instead of local score perturbations. Whether end-to-end methods can still learn more robust policies on natural instance distributions remains open. A natural next step is to understand which instance families have robust policies, and whether such robustness requires information beyond local score based rules.
>
> **Q2.**
>
> Most of our constructions use classical stable set families for cuts and binary packing families for branching. They are, by construction, worst case examples and we do not view them as direct models of average solver behavior. Their purpose is to isolate a mechanism: local signals can be misaligned with global tree size, and small local changes can be amplified by the branch-and-bound recursion. For this reason the constructions are simplified, but the mechanism they isolate is consistent with recent theory and experiments. For example, Shah et al. [1] show that stronger relaxations can increase tree size, and in their experiments they find that when the gap closed by cuts is small, the change in tree size is hard to predict and can increase. Recent learning based work on node and cut selection also moves beyond purely local information by using richer tree representations or broader cut context [4,5].
>
> Thus, we view these results as another reason to study end-to-end learning and other approaches that optimize tree size or solve effort more directly, including recent MDP based approaches [2,3]. The main lesson is that local score accuracy alone is not enough evidence of good global performance. On a given benchmark family, this can be tested by measuring how sensitive tree size or solve effort is to small score perturbations or tie-breaking changes for learned branch-and-cut policies.
>
> **References.**
>
> [1] Shah et al. (2025), Non-Monotonicity of Branching Rules with Respect to Linear Relaxations.
>
> [2] Scavuzzo et al. (2022), Learning to Branch with Tree MDPs.
>
> [3] Strang et al. (2025), A Markov Decision Process for Variable Selection in Branch & Bound.
>
> [4] Zhang et al. (2025), Learning to Select Nodes in Branch and Bound with Sufficient Tree Representation.
>
> [5] Zeng et al. (2025), Beyond Local Selection: Global Cut Selection for Enhanced Mixed-Integer Programming.

---

> > ### Author Rebuttal · Reviewer_EYY5 · 2026-03-31
> >
> > The authors provided thoughtful responses to my comments.

---

> > > ### Author Response · Authors · 2026-04-01
> > >
> > > Thank you for the thoughtful review and follow-up. We appreciate your time and feedback.

---

### Official Review · Reviewer_oVND · 2026-03-13

**Soundness:** 3
**Presentation:** 4
**Significance:** 3
**Originality:** 4
**Overall Recommendation:** 5
**Confidence:** 4

**Summary:**

This study identifies intrinsic issues within the Branch-and-Cut (B\&C) framework that arise when decision-making, such as cutting plane selection or branching decisions, relies solely on local signals.
The authors point out the limitations of existing approaches that train agents or policies using these local signals.
Specifically, they demonstrate that training a policy with computationally expensive 1-step look-ahead signals (e.g., LP improvement) can paradoxically result in larger Branch-and-Bound (B\&B) tree sizes compared to using cheaper proxy measures.
Furthermore, the study utilizes concrete examples, such as strong branching and cut set perturbation, to illustrate the inherent difficulty of learning effective B\&C decision policies based on purely local information.

**Compliance With Llm Reviewing Policy:**

Affirmed.

**Key Questions For Authors:**

1. Regarding the concerns raised above about the lack of generality, could you properly address the possibility that policies trained solely on expensive local signals might actually improve performance in certain problem classes?

2. In most modern MIP solvers and problem-solving strategies, RHS values are often made discrete to avoid minor rounding issues. How would the claims in Theorem 4.3 be modified or improved for such situations?

3. To strengthen your theoretical arguments, do you have preliminary empirical results, or plans to test a simple end-to-end learning strategy, demonstrating that global objectives outperform local-signal policies in practice?

**Limitations:**

Yes

**Strengths And Weaknesses:**

- The issues identified by the authors have been consistently raised, both empirically and theoretically, within the Branch-and-Cut (B\&C) literature, and it is certain that their research provides insights into understanding these phenomena.
I am in full agreement with their fundamental arguments.

- However, the specific instances and examples provided to support their claims feel somewhat lacking in generality.
One could potentially counter their argument by presenting specific instances where policies trained solely on expensive local signals, in a manner similar to the authors' approach, actually lead to better problem-solving performance.

- Furthermore, in Theorem 4.3, the authors argue that subtle perturbations in the Right-Hand Side (RHS) can alter the dominance of cut sets, potentially resulting in an exponential increase in the size of the Branch-and-Bound (B\&B) tree.
However, I think that these claims might be irrelevant for problem instances in pure Integer Programming (IP) or Combinatorial Optimization.
For example, in the case of Set Packing Polyhedra, consider the facet-defining inequalities: odd-cycle inequalities and clique inequalities.
The RHS of the former varies with the cardinality of the corresponding set, while the latter is fixed at 1.
This implies that when the cut family changes, the RHS shifts by a significant margin, suggesting that small perturbations on the RHS are trivial.
Moreover, in pure IP, rounding errors in the RHS of an inequality can be trivial due to integrality.
Even if the scenarios the authors are concerned about were to occur, I think that modern optimization solvers are capable of preemptively mitigating errors caused by such perturbations to a certain extent.

- The submitted paper is well-structured, with a balanced distribution of content across all sections.
In particular, in Section 4, following the technical and theoretical descriptions of each proposed methodology with an explanation of its significance and intuition was highly effective.
This approach significantly aided in clarifying the author's intent.


- The topics they tackled have been significant issues in the literature.
The authors attempted to interpret issues consistently observed in the literature through theoretical arguments.
While the act of solidifying our empirical and intuitive knowledge with theoretical statements should by no means be undervalued, it remains difficult to conclude that their claims and descriptions have significantly advanced the overall understanding of the field.


- By providing a theoretical interpretation of the previously observed intrinsic problems within the B\&C framework, they argue that policies should be trained based on global objectives rather than local signals.
This claim is valid to some extent.
However, the overall argument would have been more complete had they presented even preliminary results by implementing a simple version of an end-to-end learning strategy.

---

> ### Author Rebuttal · Authors · 2026-03-30
>
> Thank you for the positive assessment and thoughtful questions. We are glad that you find the paper well structured and agree with the fundamental arguments.
>
> **Q1.**
>
> Yes, on structured problem classes, such policies can certainly have empirical success. In fact, we are not against local imitation, and we still think such techniques can and should be deployed, but with careful evaluation and stress testing. As you note, related concerns have already been raised both empirically and theoretically in the B&C literature. For example, Ye et al. [1] note that "one wrong decision may cause a doubled tree size." Our goal in this paper is to provide *theoretical foundations* for these observations and insight into why and how such undesirable behavior can happen, so that we can focus on methods and evaluation policies to mitigate them.
>
> Further, our constructions show that the "bad" behavior of local signals and learned policies based on such signals can be quite catastrophic in the worst case, with exponential blowup in tree size. This amplifies the concerns raised by empirical work, where the observed performance degradation might appear to be not overly severe.
>
> **Q2.**
>
> We do not have stronger combinatorial sensitivity results in the paper. Extending the analysis to combinatorial cutting plane families would provide stronger evidence in solver settings, and we think this is a very good future research direction. We also agree that, for the triangle construction, the perturbed inequality is equivalent on integer points to the original, so standard preprocessing can recover the tighter cut. We therefore do not claim that this exact perturbation would survive unchanged in a modern pure IP pipeline.
>
> However, we believe the result remains relevant to IP for three reasons.
>
> 1. At the formulation level, Theorem 4.3 can be viewed as an RHS sensitivity result for branch-and-bound tree complexity, illustrating the importance of IP formulations for branch-and-cut performance. This also connects to recent work showing that tighter relaxations can increase tree size exponentially [2].
>
> 2. Remark 4.4 shows that the same construction can also be read as a sensitivity result for cut selection scores: a small perturbation or estimation error in the score used to rank candidate cuts can reverse their ordering and trigger the same exponential gap.
>
> 3. Remark 4.5 shows that a cut set can close an arbitrarily large fraction of the integrality gap and still yield an exponentially larger tree, so near complete gap closure alone does not control tree size. This also contrasts with the empirical intuition discussed in Shah et al. [2].
>
> More broadly, many classical cut families such as GMI and CG cuts, as well as more modern multirow families based on cut generating functions, still involve continuous scores in both solver implementations and machine learning based selection pipelines [3,4,5,6]. Even when the underlying problem data are rational, standard solver implementations compute and normalize such cuts in floating point, and numerical safety steps can modify coefficients and relax the cut RHS to ensure validity of the cut [7,8]. The amplification mechanism highlighted by Theorem 4.3 therefore remains relevant.
>
> **Q3.**
>
> We do not yet have preliminary empirical results of this kind in the current submission. We agree that such evidence would strengthen the paper. A natural next step is to compare a simple end-to-end strategy trained on a global objective such as tree size or solve effort against local-score imitation, first on our constructed families as a stress test and then on standard benchmarks, in line with existing work on globally aligned MDP and RL formulations [9,10].
>
> **References.**
>
> [1] Ye et al. (2023), GNN&GBDT-Guided Fast Optimizing Framework for Large-Scale Integer Programming.
>
> [2] Shah et al. (2025), Non-Monotonicity of Branching Rules with Respect to Linear Relaxations.
>
> [3] Cheng and Basu (2024), Learning Cut Generating Functions for Integer Programming.
>
> [4] Huang et al. (2022), Learning to Select Cuts for Efficient Mixed-Integer Programming.
>
> [5] Cheng et al. (2024), Sample Complexity of Algorithm Selection Using Neural Networks and Its Applications to Branch-and-Cut.
>
> [6] Balcan et al. (2022), Structural Analysis of Branch-and-Cut and the Learnability of Gomory Mixed Integer Cuts.
>
> [7] Cook et al. (2009), Numerically Safe Gomory Mixed-Integer Cuts.
>
> [8] Cornuéjols et al. (2013), On the Safety of Gomory Cut Generators.
>
> [9] Scavuzzo et al. (2022), Learning to Branch with Tree MDPs.
>
> [10] Strang et al. (2025), A Markov Decision Process for Variable Selection in Branch & Bound.

---

> > ### Author Rebuttal · Reviewer_oVND · 2026-04-04
> >
> > Thanks for the update. My questions are fully addressed.

---

> > > ### Author Response · Authors · 2026-04-04
> > >
> > > Thank you again for your thoughtful review, and we are glad that the clarifications were helpful.

---

### Official Review · Reviewer_nBcc · 2026-03-13

**Soundness:** 3
**Presentation:** 3
**Significance:** 3
**Originality:** 3
**Overall Recommendation:** 5
**Confidence:** 3

**Summary:**

This paper studies a central question in machine learning for branch and cut: whether accurately imitating local expert signals, such as strong branching scores for branching or LP bound improvement for cut selection, is enough to ensure similar global search performance. The paper answers this question in the negative through a series of worst case separations showing that local supervision can be badly misaligned with global tree size. On the cut selection side, the analysis shows that LP improvement based selection can lead to exponentially larger trees than a simpler proxy based rule, and that arbitrarily small perturbations in the right hand sides of root cuts can drastically change the resulting tree size. On the branching side, the paper shows that arbitrarily small score discrepancies, tie breaking differences, or only a small number of deviations from strong branching can already produce exponential blowups in tree size. Overall, the paper positions these results as evidence that local score accuracy alone is not a reliable surrogate for global branch and cut performance, and uses this perspective to motivate training and evaluation procedures that are better aligned with tree size and with stability under perturbations.

**Compliance With Llm Reviewing Policy:**

Affirmed.

**Key Questions For Authors:**

1. Can the paper include a small empirical stress test on existing learned branching policies? Even a limited experiment would help connect the theoretical failure modes to practice and broaden the paper’s relevance for a machine learning audience.

2. The paper’s main novelty appears to lie in the instability and cut selection separations, while part of the motivation relies on earlier results about the suboptimality of strong branching. This distinction could be stated more clearly so that the central contribution is easier to identify.

3. The practical relevance of the constructions could be discussed more directly. How much do the conclusions depend on root cut analysis, specific abstractions, or idealized branching settings in comparison with modern solver pipelines?

4. The impact discussion could be sharpened by stating more concretely what methodological lessons follow for training objectives, evaluation protocols, and stress testing of learned solver policies.

**Limitations:**

Yes

**Strengths And Weaknesses:**

**Strengths.**

Rather than proposing another learning rule, the paper identifies structural obstacles that affect a broad family of pipelines based on local objectives, which gives the work a clearer point of departure from prior papers focused mainly on incremental predictive improvements. The main results are compelling because the separations arise from small perturbations, tie breaking differences, or only a few decision deviations. Findings of this kind have clear implications for the evaluation of learned solver policies, since they show that local agreement can remain consistent with large downstream differences in search behavior. The results are also presented with appropriate care in scope, as worst case statements, and the discussion does not overextend them into claims about practical failure of all local supervision methods.

**Weaknesses.**

Part of the motivation builds on prior work showing that strong branching itself can be exponentially suboptimal. The strongest novelty here appears to lie in the instability and cut selection separations, and that distinction could be highlighted more clearly so that the paper’s specific contribution stands out more sharply. The practical relevance of some constructions to modern solver pipelines could also be discussed in more detail. In particular, it would help to clarify how much the conclusions depend on root cut analysis, specific abstractions, or idealized branching settings, since these choices affect how directly the results transfer to contemporary practice. The impact discussion could also be developed further. Since much of the paper’s value lies in methodological guidance for a fast growing area, a fuller discussion of what these results imply for evaluation protocols, benchmark design, and future learning objectives would strengthen the broader significance of the work.

---

> ### Author Rebuttal · Authors · 2026-03-30
>
> Thank you for the thoughtful review and for finding the paper technically solid.
>
> **Q1.**
> We agree that an empirical bridge is valuable. We have run preliminary computational checks on the families of instances constructed in the paper, and they are consistent with the instability phenomena outlined in the paper for both noisy strong branching and cut perturbations. A broader next step is to test whether analogous sensitivity occurs for learned policies on standard MILP benchmark families such as MIPLIB.
>
> **Q2.**
> We summarized this in Table 1, and we will make it more explicit in the introduction.
>
> **Q3.**
>
> 1. For branching, our standing assumptions follow the standard strong-branching setup in Dey et al. [1] and related theoretical analyses of branch-and-cut [2,3]. Within this model, our proofs show that if certain "wrong" variables are selected for branching (even at relatively small number of nodes) this can have catastrophic effects. We believe these wrong choices can be made even under real solver conditions. In that sense, we suspect that the instabilities caused due to the branching rules are not artificial phenomena arising out of contrived theoretical settings, but will occur independent of specific solver settings and computing environments.
>
> 2. For cut selection, our focus on root cuts is deliberate: LP bound improvement is typically computed and used there, and much of the prior learning-to-cut literature studies exactly this regime [4,5,6,7,8]. Moreover, our core cut instability result, Theorem 4.3, is about optimal sized trees: an $\varepsilon$ perturbation of a root cut set can change the optimal tree size (over all branch-and-bound trees) from $1$ to $2^{\Omega(n)}$. Thus, the conclusion is not an artifact of a particular downstream branching rule.
>
> Our point is that the practically central regimes studied here already isolate a genuine local/global misalignment mechanism. Related sensitivity phenomena have also been observed in practice. For example, Ye et al. [9] remark that "one wrong decision may cause a doubled tree size," and Reviewer oVND also notes that related issues have been consistently raised in the B&C literature.
>
> **Q4.**
>
> 1. Using training objectives or proxies aligned more directly with global performance, such as tree size or solve effort, rather than only mimicking local expert scores. MDP and RL formulations [10,11] move in this direction.
> 2. Evaluating and stress-testing learned policies under explicit perturbation models, not only by average test performance: for example, perturb predicted scores or cut definitions within a small budget, randomize tie-breaking, or force a small number of alternative decisions, and report the resulting tree-size degradation. This is analogous in spirit to robust learning/training against a specified perturbation set [12] and robustness evaluation under an explicit threat model [13], similar to adversarial robustness work in deep learning.
>
> This will help to distinguish training failures/successes from the intrinsic sensitivity of B&C recursion.
>
> **References.**
>
> [1] Dey et al. (2024), A Theoretical and Computational Analysis of Full Strong-Branching.
>
> [2] Balcan et al. (2022), Structural Analysis of Branch-and-Cut and the Learnability of Gomory Mixed Integer Cuts.
>
> [3] Basu et al. (2020), Complexity of Branch-and-Bound and Cutting Planes in Mixed-Integer Optimization.
>
> [4] Paulus et al. (2022), Learning to Cut by Looking Ahead: Cutting Plane Selection via Imitation Learning.
>
> [5] Cheng et al. (2024), Sample Complexity of Algorithm Selection Using Neural Networks and Its Applications to Branch-and-Cut.
>
> [6] Cheng and Basu (2024), Learning Cut Generating Functions for Integer Programming.
>
> [7] Balcan et al. (2021), Sample Complexity of Tree Search Configuration: Cutting Planes and Beyond.
>
> [8] Kazachkov et al. (2024), An Abstract Model for Branch and Cut.
>
> [9] Ye et al. (2023), GNN&GBDT-Guided Fast Optimizing Framework for Large-Scale Integer Programming.
>
> [10] Scavuzzo et al. (2022), Learning to Branch with Tree MDPs.
>
> [11] Strang et al. (2025), A Markov Decision Process for Variable Selection in Branch & Bound.
>
> [12] Madry et al. (2018), Towards Deep Learning Models Resistant to Adversarial Attacks.
>
> [13] Carlini et al. (2019), On Evaluating Adversarial Robustness.

---

> > ### Author Rebuttal · Reviewer_nBcc · 2026-03-31
> >
> > Thank you for your responses. Your clarifications were very helpful. I will increase my score. Nice work!

---

> > > ### Author Response · Authors · 2026-04-01
> > >
> > > Thank you for the kind feedback and for taking the time to reconsider the score. We are glad that the clarifications were helpful.

---

### Decision · Program_Chairs · 2026-04-30

**Decision:**

Accept (regular)

**Comment:**

This paper examines the limitations of current machine learning models used to guide branch-and-cut strategies in mixed-integer linear programming (MILP) problems. The authors demonstrate that relying on local score-based models can result in exponential growth of the search tree. This work highlights the need for alternative approaches that are more closely aligned with controlling search tree size.

All reviewers have given a unanimous score of 5, emphasizing the paper's originality, clear structure, and significant implications for future research in learning-based MILP solving. Based on these strengths, I recommend acceptance.